# Coral endosymbiont growth is enhanced by metabolic interactions with bacteria

Jennifer L. Matthews [1] ✉, Abeeha Khalil [1], Nachshon Siboni [1], Jeremy Bougoure[2], Paul Guagliardo [2], Unnikrishnan Kuzhiumparambil[1], Matthew DeMaere [1], Nine M. Le Reun [1], Justin R. Seymour [1], David J. Suggett [1,3] & Jean-Baptiste Raina [1]

Bacteria are key contributors to microalgae resource acquisition, competitive performance, and functional diversity, but their potential metabolic interactions with coral microalgal endosymbionts (Symbiodiniaceae) have been largely overlooked. Here, we show that altering the bacterial composition of two widespread Symbiodiniaceae species, during their free-living stage, results in a significant shift in their cellular metabolism. Indeed, the abundance of monosaccharides and the key phytohormone indole-3-acetic acid (IAA) were correlated with the presence of specific bacteria, including members of the *Labrenzia* (*Roseibium*) and *Marinobacter* genera. Single-cell stable isotope tracking revealed that these two bacterial genera are involved in reciprocal exchanges of carbon and nitrogen with Symbiodiniaceae. We identified the provision of IAA by *Labrenzia* and *Marinobacter*, and this metabolite caused a significant growth enhancement of Symbiodiniaceae. By unravelling these interkingdom interactions, our work demonstrates how specific bacterial associates fundamentally govern Symbiodiniaceae fitness.

Metabolic exchanges between autotrophic organisms and heterotrophic prokaryotes (i.e., bacteria and archaea) underpin the primary production and ecological success of most plants and algae across marine and terrestrial systems[1,2]. Microalgal hosts release compounds that can enhance the growth of specific bacterial associates while inhibiting the growth of other unwanted microbes[3,4]. In return, bacterial partners release various stimulatory compounds promoting the growth of their algal hosts, such as siderophores, B-vitamins, hormones[5–7], and signal molecules (e.g., alkylquinolone quorum-sensing system), which facilitate coordinated and cooperative behaviours that are biogeochemically important[8,9], and may govern evolutionary relationships between phytoplankton and their associated microbiome[10].

Microalgal dinoflagellates of the family Symbiodiniaceae[11] are important primary producers in the global ocean[12] and their endosymbiosis with reef-building corals underpins reef accretion and the health and productivity of coral reefs[13]. Cultured Symbiodiniaceae can predate on bacteria under both nutrient-replete and -depleted conditions, demonstrating that heterotrophic feeding may be a survival strategy for Symbiodiniaceae cells when free-living[14]. However, dependency of free-living Symbiodiniaceae upon bacteria for resource acquisition likely extends beyond predation to involve the mutual exchange of key metabolites[3,14–18]. For example, Symbiodiniaceae cultures harbour a core microbiome (including *Labrenzia*, *Marinobacter*, and Chromatiaceae)[18–20], as well as surface associated and highly conserved intracellular bacteria[21]. Volatile compounds emitted by Symbiodiniaceae-associated bacteria are modulated by the presence of Symbiodiniaceae exudates, suggesting complex chemical interplay between these microalgae and their associated bacteria[22], potentially creating a selective phycosphere (i.e., the microenvironment enriched in photosynthates surrounding the microalgal cells[3,23]). Finally, bacterial communities associated with heat-evolved strains of

[1]Climate Change Cluster, University of Technology Sydney, Ultimo, NSW 2007, Australia. [2]Centre for Microscopy, Characterisation and Analysis, University of Western Australia, Perth, WA 6009, Australia. [3]Present address: KAUST Reefscape Restoration Initiative (KRRI) and Red Sea Research Center (RSRC), King Abdullah University of Science and Technology, Thuwal 23955, Saudi Arabia. ✉e-mail: Jennifer.Matthews@uts.edu.au

*Cladocopium* are more stable than wild-type strains[20], and specific bacteria can produce the potent antioxidant zeaxanthin[16], suggesting a functional role for these bacteria in Symbiodiniaceae environmental resilience[19]. Collectively such evidence, combined with historical difficulty in growing Symbiodiniaceae axenically[24], suggests important biological links between Symbiodiniaceae and their bacterial partners[15,19,24,25].

Despite the importance of microalgal-bacteria interactions in primary production and nutrient cycling throughout marine environments[3], the role of bacteria in Symbiodiniaceae health, resource acquisition, and competitive performance remains a fundamental gap in knowledge that wholly constrains our understanding of how microbes act in concert to regulate the health of their coral hosts[15,26–30]. Exploring Symbiodiniaceae-bacteria metabolic associations is therefore a crucial step towards fully understanding the complex symbiotic interactions occurring in the coral holobiont[15]. Here we applied an integrated microbiome and metabolite profiling approach to identify the chemical currencies involved in these partnerships, and explore whether widespread Symbiodiniaceae species hold an innate

dependency on bacteria for resource exchange in their free-living state, thus forming intimate associations that may support optimum metabolic fitness of the coral holobiont. Our results reveal that modifying the bacterial communities of cultured *Symbiodinium microadriaticum* and *Breviolum minutum* significantly affects their metabolite profiles, and that the bacterial associates *Labrenzia alexandrii* and *Marinobacter* sp. produce the key phytohormone indole-3-acetic acid (IAA), which enhances the growth of these two Symbiodiniaceae species.

## Results and Discussion

To identify potential metabolic interactions and interdependencies between Symbiodiniaceae and specific bacterial associates, we initially altered the bacterial community composition of two Symbiodiniaceae species – *Symbiodinium microadriaticum* and *Breviolum minutum* – via a single combination of detergent (Triton-X100) and antibiotic treatment (Ab+Tx, Supplementary Data 1, Fig S1), and examined the effects of these alterations on the photophysiology and metabolite profiles of *S. microadriaticum* and *B. minutum* (Supplementary Data 2-7, Fig. 1, S2).

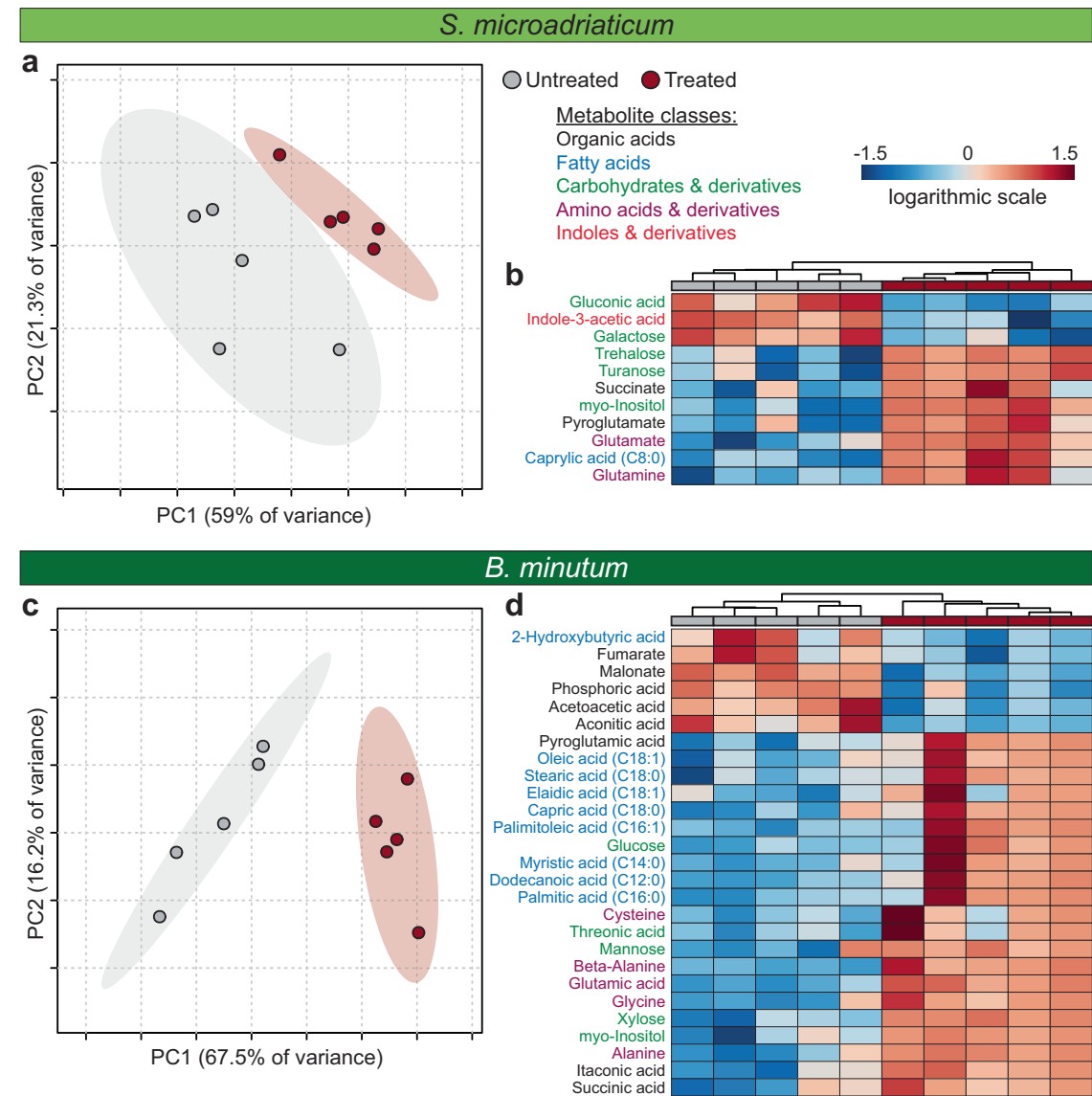

**Fig. 1 | Effect of antibiotic treatment on Symbiodiniaceae metabolite profiles.**
PCA plots of metabolite profiles (**a**, **c**) and heatmaps of the significantly different metabolites (**b**, **d**; One way ANOVA, FDR < 0.05) for cultures of untreated (grey) and Ab+Tx treated (maroon) *Symbiodinium microadriaticum* (**a**, **b**) and *Breviolum minutum* (**c**, **d**) (*n* = 5 per treatment, per species). Metabolite names are coloured according to class; organic acids (black), fatty acids (blue), carbohydrates and derivatives (green), amino acids (purple), and indoles and derivatives (red). Source data are provided as a Source Data file.

Ab+Tx treatment significantly reduced the total number of bacteria cells per Symbiodiniaceae by 1.19 times in *S. microadriaticum* and 2.60 times in *B. minutum*, and altered the microbial profile of both Symbiodiniaceae species (PERMANOVA, *p* < 0.001; Supplementary Data 2 and 4, Fig S1). A total of 12 and 13 bacterial genera significantly decreased in relative abundance for treated *S. microadriaticum* and *B. minutum*, respectively, including *Labrenzia (Roseibium)*, *Marinobacter* and *Muricauda* (Mann Whitney U, Supplementary Data 6). Seven days after this treatment, cell photophysiology remained unchanged for both *S. microadriaticum* and *B. minutum* (Fig. S2, Supplementary Data 7), but the specific growth rates of *S. microadriaticum* and *B. minutum* were reduced by alteration in the bacterial consortium, by 86.2% and 8.2%, respectively (ANOVA, *p* < 0.001, Supplementary Data 8), implying growth rate dependence on healthy bacterial consortia. Consistent with the growth response, marked alterations were observed in the free metabolite pools of each species (Fig. 1a-d, Supplementary Data 9-10). For *S. microadriaticum* and *B. minutum*, 7 and 28 metabolites significantly increased in relative abundance in response to Ab+Tx treatment respectively, including carbohydrates, *myo*-inositol, and precursors of the antioxidant glutathione (T-test, FDR < 0.05; Fig. 1b, d, Supplementary Data 10). Meanwhile, 3 and 6 metabolites significantly decreased in *S. microadriaticum* and *B. minutum*, respectively, including the growth-promoting hormone indole-3-acetic acid (IAA), fatty acid metabolism by-products and TCA cycle intermediates (T-test, FDR < 0.05; Fig. 1b, d, Supplementary Data 10).

Using correlative analysis, we aimed to identify candidate metabolites-bacteria combinations that differed between the untreated and treated Symbiodiniaceae cultures. Treated and untreated cultures for both *S. microadriaticum* (Fig. 2a, c, d) and *B. minutum* (Fig. 2b, e, f) were distinct in their microbe-metabolite pairings (PLS-DA, *p* < 0.001). For *S. microadriaticum*, 13 metabolites were positively correlated with 21 bacteria genera, with 69% of the metabolites and 25% of the bacterial taxa increasing in relative abundance in treated cultures (Spearman's correlation coefficient (*r*); Fig. 2a, c, d, Supplementary Data 11). For *B. minutum*, 28 metabolites were positively correlated with 17 bacteria genera, of which 79% of the metabolites and 24% of the bacteria taxa increased in relative abundance in treated cultures (Spearman's correlation coefficient (*r*); Fig. 2b, e, f, Supplementary Data 11). These microbe-metabolite pairings constitute a rich dataset to guide our in-depth characterisations of the functional roles of these bacteria for Symbiodiniaceae. For example, cell-surface bound mannose, xylose and galactose are involved in glycan-lectin interactions and the establishment of the cnidarian-Symbiodiniaceae symbiosis and the recognition of beneficial bacteria symbionts by corals[31–38]. Therefore, the positive correlation of mannose and xylose with *Muricauda* and galactose with *Labrenzia*, both of which are previously described core microbiome members and intracellular symbionts of Symbiodiniaceae[17–19,21], may indicate potential interkingdom signalling mechanisms with these Symbiodiniaceae species. Furthermore, several metabolites previously involved in Symbiodiniaceae chemical signalling (e.g. trehalose), stress protection (e.g. glutathione and *myo*-inositol), Symbiodiniaceae or bacteria nutrition (e.g. glucose, β-alanine), and microalgal auxins (e.g. indole-3 acetic acid) were also significantly correlated with microbiome members of Symbiodiniaceae (e.g. *Labrenzia, Muricauda, Pseudomonas*); a detailed description of these putative interactions are provided in the Table 1.

To identify specific metabolite exchanges between Symbiodiniaceae and their associated bacteria, we first successfully isolated 7 species of bacteria from untreated *S. microadriaticum* and *Breviolum minutum* cultures, including strains with more than 98% sequence identity with *Labrenzia alexandrii, Marinobacter* sp., and *Muricauda aquimarina* (Supplementary Data 12). These three bacterial genera are members of the Symbiodiniaceae core microbiome[18,21] and may have important functions for these dinoflagellates, including the potential

translocation of antioxidant compounds[16]. Given the ubiquitous presence of *Labrenzia, Marinobacter*, and *Muricauda* across Symbiodiniaceae genera[18] as well as their strong correlation with multiple chemicals identified here, we quantified metabolic exchanges between members of these genera and Symbiodiniaceae using nanoscale secondary ion mass spectrometry (NanoSIMS). We completely removed extracellular bacteria from *S. microadriaticum* and *B. minutum* cultures via 4 rounds of filtering, rinsing with 6% sodium hypochlorite and 48 h of incubation with antibiotics (Supplementary Data 1), resulting in cultures we henceforth termed "extracellular bacteria-removed" (EBR; Supplementary Data 13), making them suitable for co-culture experiments with individual bacterial isolates. Prior to co-culturing, the three bacterial strains (*L. alexandrii, Marinobacter* sp., and *M. aquimarina*; Supplementary Data 12) were pre-enriched with $^{15}$N, and the EBR cultures of the two Symbiodiniaceae species were pre-enriched with the rare stable isotope $^{13}$C.

After 6 hours of co-culturing, all three bacterial strains were significantly enriched in $^{13}$C derived from Symbiodiniaceae (Fig. 3, Supplementary Data 14, ANOVA; FDR < 0.001). Following co-culturing, $^{13}$C enrichment of *L. alexandrii* cells was on average 2.7 times greater than natural abundance levels (Fig. 3, Supplementary Data 13; ANOVA, FDR < 0.001), and very similar following co-culture with both Symbiodiniaceae species (Fig. 3, Supplementary Data 14 ANOVA; FDR = 0.062). In comparison, both *Marinobacter* sp. and *M. aquimarina* were more enriched when associated with *S. microadriaticum*, from which the bacteria were originally isolated (Fig. 3, Supplementary Data 12&S14; ANOVA, FDR < 0.001), with $^{13}$C levels on average 5 times higher than the natural abundance in *M. aquimarina* (Fig. 3, Supplementary Data 14). Surprisingly, both Symbiodiniaceae species were significantly enriched in $^{15}$N derived from *L. alexandrii* and *Marinobacter* sp. (Fig. 3, Supplementary Data 14; ANOVA, FDR < 0.001), but not from *M. aquimarina* (Fig. 3, Supplementary Data 14; ANOVA, *S. microadriaticum:* FDR = 0.704, *B. minutum:* FDR = 0.438). Indeed, *S. microadriaticum* and *B. minutum* $^{15}$N levels were on average 2.62 and 2.95 times greater than natural abundance when associated with *L. alexandrii*, and 1.82 and 3.69 times greater than natural abundance when associated with *Marinobacter* sp., respectively. These results reveal that reciprocal chemical exchange occurs between Symbiodiniaceae and some of their associated bacteria.

EBR cultures of both Symbiodiniaceae species were next co-cultured with the *L. alexandrii* and *Marinobacter* sp. strains to examine the consequences of these reciprocal metabolic exchanges. Both bacteria enhanced the growth of each Symbiodiniaceae strain compared to the controls based on chlorophyll *a* fluorescence (ANOVA; FDR < 0.01, Fig. 4a, Supplementary Data 8), and each bacterium remained in culture and substantially increased in abundance during the 14 days (8.7 times more *Labrenzia* and 12.2 more *Marinobacter* per Symbiodiniaceae cells; Supplementary Data 15). Indeed, after 14 days in co-culture, *S. microadriaticum* growth increased by 80.9% in the presence of *L. alexandrii* and 81.4% in the presence of *Marinobacter* sp. relative to the EBR control cells. Similarly, *B. minutum* growth increased by 31.3% in the presence of *L. alexandrii*, and 30.2% in the presence of *Marinobacter* sp. compared to the control (Fig. 4a, Supplementary Data 8). Notably, both *Labrenzia* and *Marinobacter* are core bacteria genera across Symbiodiniaceae[18], indicating these beneficial metabolic interactions leading to growth enhancement could be present across other Symbiodiniaceae species.

To further identify the bacterial metabolites driving the Symbiodiniaceae growth enhancements, we used the correlation data to select metabolites connected to these bacteria with the highest *r* values (and thus strongest interaction). In *S. microadriaticum* cultures, the correlation between indole-3-acetic acid (IAA) and the genus *Labrenzia* had the largest *r* value and highest significance (*r* = 0.964, *p* < 0.001; Supplementary Data 11), and IAA was the only metabolite significantly correlated with *Marinobacter* sp. (*r* = 0.709, *p* = 0.002, Supplementary

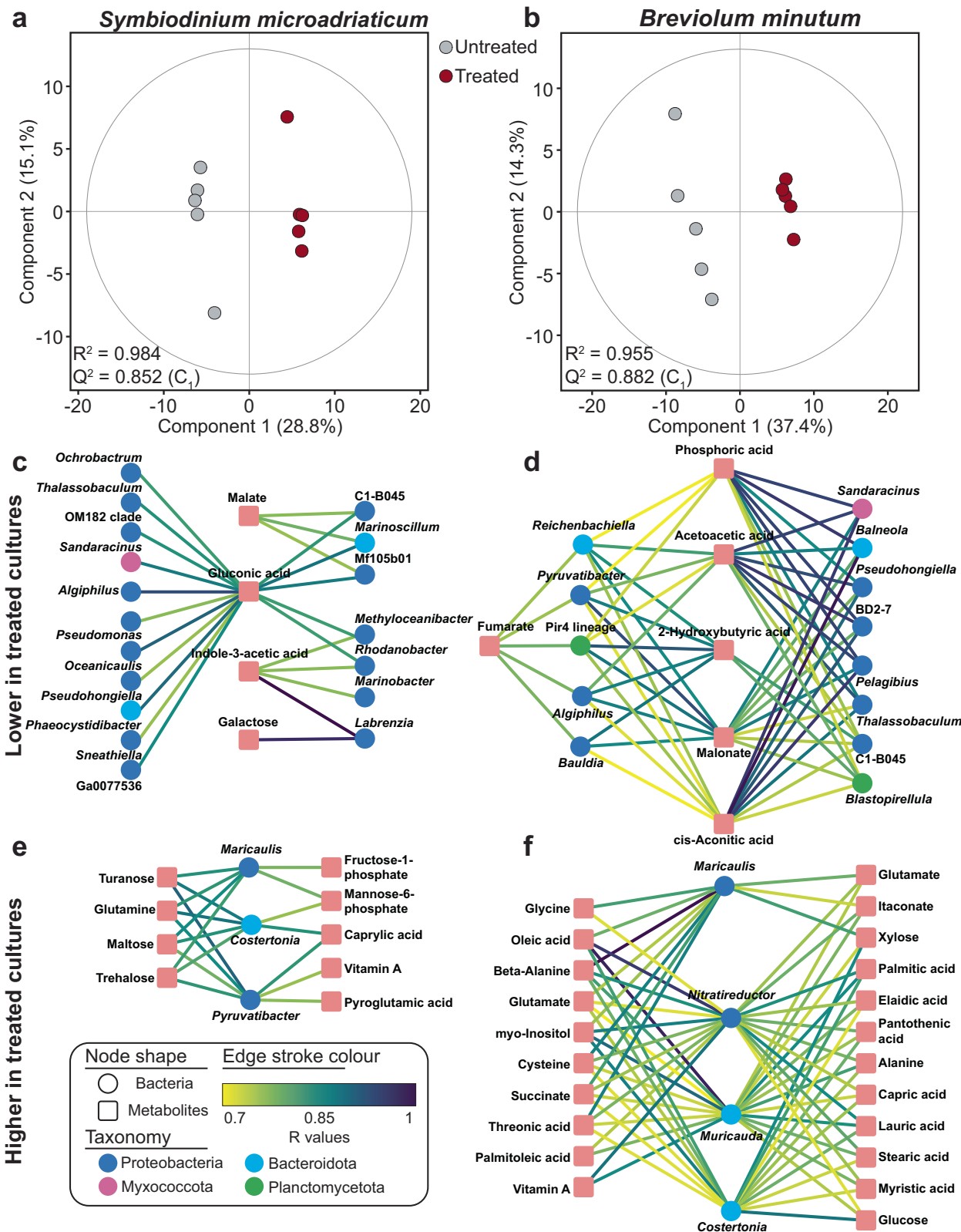

Data 11). IAA is a phytohormone that enhances the growth of plants and recent evidence suggests that it can also promote the growth of microalgae belonging to chlorophytes and diatoms[5,39,40]. It is synthesised and released by bacteria from multiple classes (including Bacilli, Alpha-, and Gamma-proteobacteria[41]) most commonly using tryptophan acquired from microalgal partners[5,42]. We first confirmed the tryptophan-dependent production of IAA using our *L. alexandrii*, and

*Marinobacter* sp. isolates (Supplementary Data 12) via LC-MS analysis (Fig. 4b). When tryptophan (final concentration 50 nM) was provided to each of the bacterial isolates during exponential growth, IAA concentration per cell significantly increased by 4.5 times in *L. alexandrii*, and 16.6 times in *Marinobacter* sp (Fig. 4b, Supplementary Data 16). Strikingly, when the bacteria were exposed to tryptophan, the IAA concentration exuded by the cells increased even more than the

**Fig. 2 | The influence of antibiotic treatment on the microbial and metabolite composition of *Symbiodinium microadriaticum* and *Breviolum minutum* cultures.** Multivariate regression analysis of the microbe-metabolite data revealed the Ab+Tx treated (maroon) and untreated cultures (grey) for (**a**) *S. microadriaticum* and (**b**) *B. minutum* were significantly distinct (PLS-DA, $p < 0.001$, $n = 5$). For *S. microadriaticum*, Spearman's correlation coefficient (*r*) identified 13 metabolites (squares) positively correlated with 21 bacteria genera (circles); for *B. minutum*, 28 metabolites were positively correlated with 17 bacteria genera. In *S. microadriaticum*, (4 metabolites and 18 bacteria significantly decreased in relative abundance in treated cultures (**c**), and 6 metabolites and 13 bacteria genera significantly decreased in relative abundance in treated *B. minutum* cultures (**d**). In comparison, 9 metabolites and 3 bacteria significantly increased in relative abundance in treated *S. microadriaticum* cultures (**e**), while 22 metabolites and 4 bacteria significantly decreased in relative abundance in treated *B. minutum* cultures (**f**) (Supplementary Data 11). Bacteria are coloured according to their taxonomy (Proteobacteria = navy, Myxococcota = purple, Bacteroidota = blue, Planctomycetota = green). Edge colours are according to R values. Source data are provided as a Source Data file.

**Table 1 | Microbe-metabolite correlations and putative interactions representing key priority targets for research**

| Bacterial genera | Metabolites | Putative Interaction | Evidence |
|---|---|---|---|
| *Muricauda* *Labrenzia* | Mannose, xylose Galactose | Interkingdom recognition and interaction | The establishment of the cnidarian-Symbiodiniaceae symbiosis and the recognition of beneficial bacteria symbionts by corals involves glycan-lectin interactions (specifically mannose-mannose, and galactose-β(1-4)-N-acetylglucosamine) and cell-surface bound mannose, xylose and galactose[31–38]. |
| *Pyruvatibacter* | Trehalose | Chemical signalling | Trehalose is a signal metabolite in plant interactions with pathogenic or symbiotic microorganism and has been identified as a chemoattractant released by Symbiodiniaceae affecting the behaviour of coral larvae[66,67]. |
| *Muricauda* *Pseudomonas* *Algiphilus* *Pseudohongiella* *Pyruvatibacter* | Glucose Xylose Pantothenic acid β-alanine Gluconic acid Turanose | Nutritional interaction | Glucose is a well characterised Symbiodiniaceae exudate and glucose and xylose excreted by microalgae can enhance microbial biomass[68–71]. β-amino acids and pantothenic acid play important roles in the regulation of nutritional metabolism and immunity, and can also be secreted to support the growth of other microbes[72,73]. Bacteria tend to use glucose as the preferential carbon source and produce gluconic acid (a carboxylic acid derivative of glucose)[74]. *Pseudomonas* has recently been identified as an intracellular core genus of Symbiodiniaceae, with *Algiphilus* as a closely associated genus and *Pseudohongiella* as a loosely associated genus[21]. Turanose is an analog of sucrose not metabolised by plants but can be acquired through sucrose transporters for intracellular carbohydrate signalling, and acts as a carbon source for numerous bacteria[75]. |
| *Muricauda* *Pyruvatibacter* | myo-Inositol Cysteine Glutamate Glutamic acid Pyroglutamic acid Glutamine | Stress protection (e.g antioxidant exchange) | In other microalgae-bacteria interactions, bacterial consortia confer microalgal tolerance to stress through the provision of micronutrients and/or horizontal gene transfer[76]. Several antioxidants known to play a key functional role in Symbiodiniaceae under abiotic stress were identified including myo-inositol, cysteine and glutathione antioxidant precursors[77–79]. *Muricauda* can release Zeaxanthin that improves photosynthetic efficiency and neutralises ROS during temperature stress[16]. *Pyruvatibacter* requires pyruvate – a well-known ROS scavenger – as a carbon source and some genomes encode genes for antioxidants such as superoxide dismutase and glutathione[80]. |

intracellular concentrations (5 times more than the controls for *L. alexandrii*, and 18.5 times more for *Marinobacter* sp.), revealing that IAA was not only produced but also exuded in large amounts by the bacterial cells (Fig. 4b, Supplementary Data 16).

To determine if bacterial-derived IAA was taken up by Symbiodiniaceae, extracellular IAA concentrations were quantified during co-culture experiments (Fig. 4d, Supplementary Data 16). Our results revealed that extracellular IAA concentrations were significantly reduced by $1.5 \times 10^4$ on average in the presence of Symbiodiniaceae cells compared to the supernatant of the non-tryptophan dosed bacteria, irrespective of the Symbiodiniaceae-species (Fig. 4d), which suggests that Symbiodiniaceae cells indeed take up bacterial-derived IAA.

IAA can be produced by some plants, although known homologs of tryptophan-dependant and independent IAA-biosynthesis genes were not detected in the published genomes of *S. microadriaticum* and *B. minutum* (Supplementary Data 17), suggesting that Symbiodiniaceae may be IAA-auxotrophs. To confirm whether bacterial derived IAA was responsible for the observed growth changes and thus acts as a Symbiodiniaceae auxin, EBR *S. microadriaticum* and *B. minutum* were incubated with 50 nM IAA, a concentration previously found to enhance the growth of the marine diatom *Pseudo-nitzschia multiseries*[5]. IAA significantly enhanced the growth rate of EBR *S. microadriaticum* by 86.4%, and the growth of EBR *B. minutum* by 7.6% (ANOVA, $p < 0.001$, Fig. 4a, Supplementary Data 8), indicating that this phytohormone is a beneficial metabolite exchanged between bacterial partners and Symbiodiniaceae. The growth enhancement was less pronounced for *B. minutum* than *S. microadriaticum*, suggesting that other bacterial-derived compounds may further enhance the growth of this Symbiodiniaceae species. While Symbiodiniaceae heterotrophic feeding on the bacteria[14] could also provide a source of IAA, this study demonstrates the capacity for IAA secretion by associated bacteria, and the up-take of exogenous IAA which enhanced the growth of Symbiodiniaceae.

Taken together, our work reveals that Symbiodiniaceae experience substantial metabolic benefits from partnerships with specific bacterial partners, and that they utilise bacteria-derived metabolites, such as the phytohormone IAA, to support their growth. We show that *L. alexandrii* and *Marinobacter* sp. are not simply conserved members of the Symbiodiniaceae microbiome, but play an active and important role in Symbiodiniaceae health and fitness. Tight exchange of essential molecules and limiting nutrients to maintain cellular functioning and mechanisms of inter-kingdom signalling is a hallmark of the cnidarian-Symbiodiniaceae symbiosis[43], but the evidence provided here also indicates an important role of bacterial assemblages in governing the health and growth of Symbiodiniaceae, and by extension the health of the coral holobiont. We are likely just scratching the surface of the chemical exchanges occurring between Symbiodiniaceae and bacteria, with many more important chemical currencies yet to be identified. Our new evidence unlocking the metabolic connections between bacteria and Symbiodiniaceae partnerships will likely prove critical in

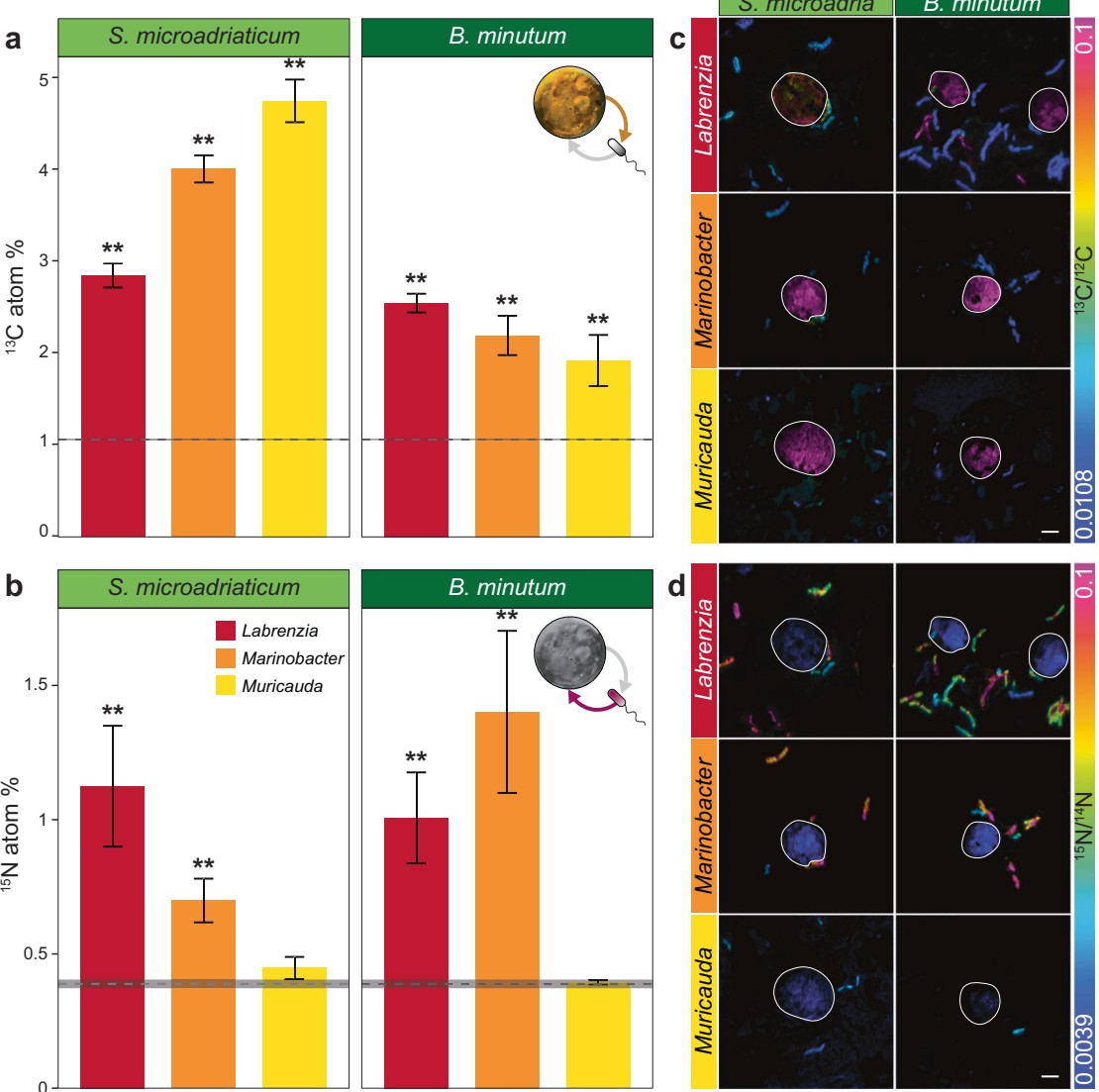

**Fig. 3 | Reciprocal exchange of metabolites in Symbiodiniaceae-bacteria co-cultures. a** *S. microadriaticum* (left) and *B. minutum* (right) were significantly more enriched in bacterial-derived [15]N when in co-culture with *Labrenzia alexandrii* (red) and *Marinobacter* sp. (orange) (One way ANOVA with Tukey's post-hoc analysis, all *p* < 0.001), but not in co-culture with *Muricauda aquimarina* (yellow) (One way ANOVA with Tukey's post-hoc analysis, *p* = 0.704 and 0.938 for *S. microadriaticum* and *B. minutum*, respectively). (One way ANOVA with Tukey's post-hoc analysis; ** = *p* < 0.001, *S. microadriaticum-L. alexandrii n* = 20; *S. microadriaticum-Marinobacter* sp. *n* = 26; *S. microadriaticum-M. aquimarina n* = 20; *B. minutum-L. alexandrii n* = 25; *B. minutum-Marinobacter* sp. *n* = 18; *B. minutum-M. aquimarina n* = 14 Symbiodiniaceae cells). Data are presented as mean values with standard error bars. Calculated abundance and statistics are detailed in Supplementary Data 14. (**b**) *L. alexandrii*, *Marinobacter* sp., and *M. aquimarina* were all significantly enriched with

Symbiodiniaceae-derived [13]C when in co-culture with both Symbiodiniaceae species. (One way ANOVA with Tukey's post hoc analysis; ** = *p* < 0.001, *S. microadriaticum-L. alexandrii n* = 93; *S. microadriaticum-Marinobacter* sp. *n* = 94; *S. microadriaticum-M. aquimarina n* = 63; *B. minutum-L. alexandrii n* = 117; *B. minutum-Marinobacter* sp. *n* = 78; *B. minutum-M. aquimarina n* = 42 bacteria cells). Error bars in **a** and **b** represent standard error of the mean, calculated abundance and statistics are detailed in Supplementary Data 14. The grey lines in **a** and **b** represent the average natural abundance in control cells (dotted line) with standard error (shading). **c–d** Representative NanoSIMS images showing the Atom % of [15]N/[14]N and [13]C/[12]C, with natural abundance in blue, changing to pink with increasing [15]N or [13]C levels. At least 20 images were obtained per co-culture treatment, per Symbiodiniaceae species. Scale bar: 3 μm. Source data are provided as a Source Data file.

ensuring the effectiveness of rapidly accelerating microbial-based reef intervention management tools[44], such as probiotics designed to mitigate against stress-induced bleaching and disease[45,46] or introduction of desired Symbiodiniaceae to host corals[47] that require optimised Symbiodiniaceae microbiomes.

## Methods

### Symbiodiniaceae cultures

Two Symbiodiniaceae cultures were targeted from existing stocks at the University of Technology Sydney, *Symbiodinium microadriaticum*

(ITS2: A1, culture ID: RT61), and *Breviolum minutum* (ITS2: B1, culture ID: RT2, CCMP2463) (Supplementary Data 18) as preliminary trials allowed these cultures to be maintained for extended periods in an extracellular bacteria-free state. Each Symbiodiniaceae species was sub-cultured (*n* = 10 per Symbiodiniaceae species) by adding 10 mL of original cultures in 90 mL of autoclaved and filter sterilised (0.22 μm) artificial seawater (ASW) and F/2 media. Cultures were grown for one month (to achieve a minimum cell density of $10^6$ cells/mL) at 26 °C with an irradiance of 85 ± 15 μmol photons m$^{-2}$ s$^{-1}$ (Philips TLD 18 W/54 fluorescent tubes, 10 000 K on a 12 h:12 h

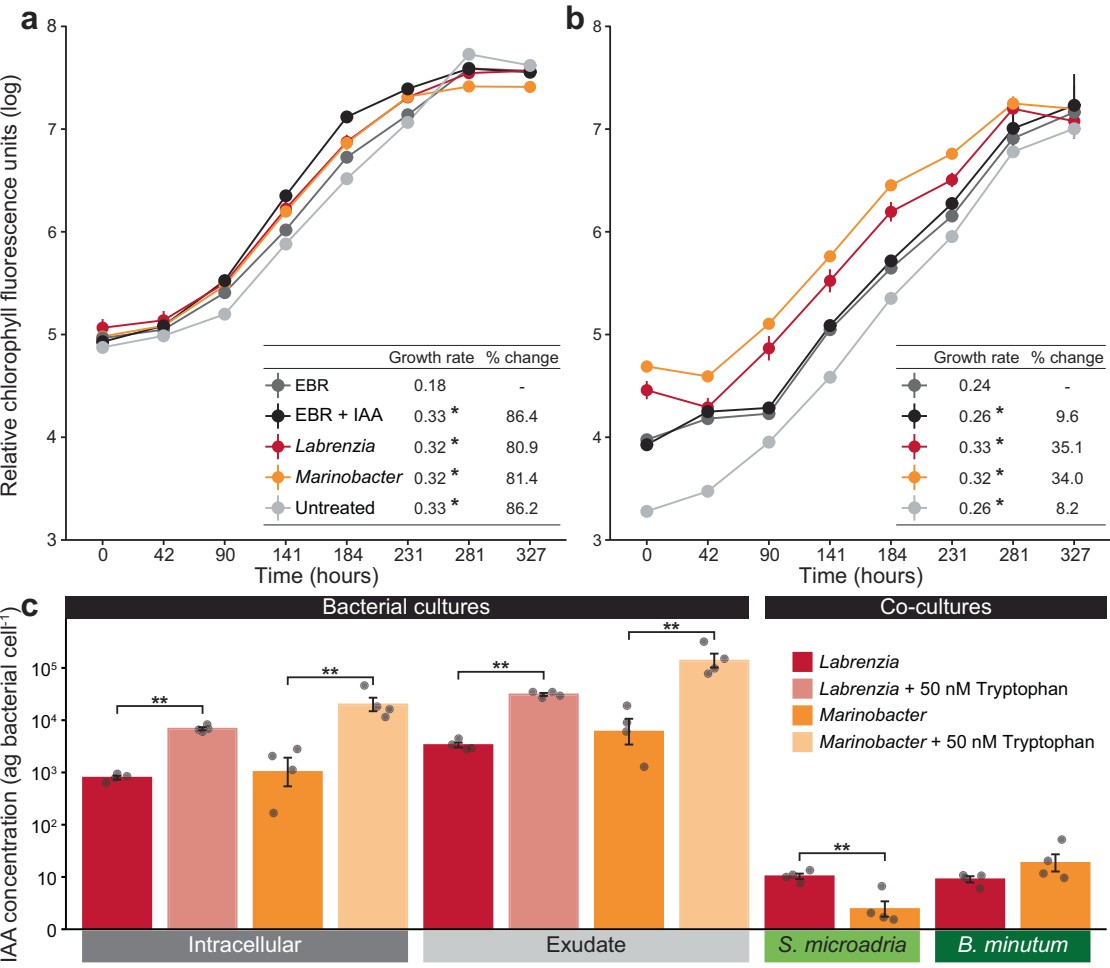

**Fig. 4 | The effect of Indole-3-acetic acid on Symbiodiniaceae and its production by *L. alexandrii* and *Marinobacter* sp. a** The growth of both *S. microadriaticum* and (**b**) *B. minutum* was significantly enhanced by the presence of IAA, or the co-culture with each bacterial species, compared to untreated and extracellular bacteria removed (EBR) cultures ($n = 5$ per treatment). **c** Quantification of the absolute abundance of IAA per bacterial cell from the intracellular and exudate fractions (in attogram per cell and displayed as a logarithmic scale). Tryptophan significantly enhanced IAA concentration for both bacteria, suggesting the tryptophan-dependant synthesis of IAA by these bacteria (Two tailed T-test, *L. alexandrii* cell extracts $t = −12.232$, df = 6, $p < 0.001$, supernatant $t = −13.607$, df = 6, $p < 0.001$, *Marinobacter* sp. cell extracts $t = −2.736$, df = 6, $p = 0.034$, supernatant $t = −2.799$, df = 6, $p = 0.031$; Supplementary Data 16, $n = 4$ per treatment, per species). When Symbiodiniaceae were in co-culture with both bacteria, the abundance of IAA in the supernatant was reduced. Moreover, there was significantly more IAA in the supernatant when *S. microadriaticum* was in co-culture with *L. alexandrii* than *Marinobacter* sp. (Two tailed T-test, $t = 4.361$, df = 6, $p = 0.005$; Supplementary Data 16, $n = 4$ per treatment), while the inverse (although not statistically significant) was true for *B. minutum* (Two tailed T-test, $t = −1.431$, df = 6, $p = 0.201$; Supplementary Data 16, $n = 4$ per treatment). Data are presented as mean values with standard error bars. Source data are provided as a Source Data file.

light:dark cycle). Before use, cells were centrifuged at $700 \times g$ for 10 mins at 26 °C and rinsed twice with ASW to remove residual media solution. Cells were resuspended in 100 mL ASW + F/2 media in sterile culture flasks.

### Antibiotic treatment (AbTx)
Each antibiotic treatment subculture ($n = 5$ per Symbiodiniaceae species) was provided with TritonX-100 detergent added to a final concentration of 20 μg/mL and placed on a shaker at mid-speed for 30 s. All cultures (Ab+Tx treatment and untreated control) were immediately centrifuged at $700 \times g$ for 10 mins at 26 °C, and the supernatant discarded. Cells were rinsed in 20 mL ASW and centrifuged at $700 \times g$ for 10 mins at 26 °C. Cells were transferred to sterile culture flasks and resuspended in 9 ml ASW + F/2. An aliquot of 1 mL custom antibiotic mix (Penicillin at 31.25 μg mL⁻¹, Streptomycin and Kanamycin at 50 μg mL⁻¹, and Neomycin, Ciprofloxacin and Ampicillin all at 100 μg mL⁻¹; Ab+Tx, Supplementary Data 1[48–51]) was added to each Ab+Tx treatment flask (and 1 mL ASW added to each untreated control), and flasks were replaced in the incubator. After 48 hours, 90 mL

ASW + F/2 was added to all cultures. Cultures were allowed to recover for 5 days prior to experimentation.

For the Extracellular Bacteria Removed (EBR) treatment, cultures ($n = 5$ per Symbiodiniaceae species) were washed with TritonX-100 as above and the detergent supernatant discarded. Cells were resuspended in 20 mL ASW, filtered on a 0.22 μm Durapore membrane filter, and rinsed with 20 mL 6% sodium hypochlorite. Cells were transferred to sterile culture flasks and resuspended in 9 ml ASW + F/2 and treated with antibiotics as above. This process was repeated four times. After 1 week, bacteria absence was confirmed by two individual tests: (1) plating 1 mL of each Symbiodiniaceae culture on marine broth solid media followed by 5 days incubation and visual checks for bacteria contamination; and (2) SYBR Green staining and absence of signal on flow cytometry following blank correction (Supplementary Data 13; Fig S3c-d; see *Symbiodiniaceae and bacterial cell density analysis* below). To further confirm bacteria absence, fixed EBR cultures (2% glutaraldehyde) were stained with 0.5 mg/mL DAPI for 5 minutes in the dark and filtered on a 0.2 μm polycarbonate black filter. Cells were then imaged under a Nikon ECLIPSE Ni-L upright fluorescence microscope

using a metal halide mercury lamp as a light source, with both DAPI filter (excitation (Ex): 359 nm; emission (Em): 457 nm) for bacteria cell visualisation and Cy5 filter (Ex: 630–650 nm; Em: 660–700 nm) to distinguish Symbiodiniaceae autofluorescence (Figure S4). Images of DAPI emission overexposed the Symbiodiniaceae cells to reveal the presence of any DNA from extracellular bacteria cells. We maintained untreated control cultures (i.e., no detergent or antibiotic treatment; $n = 5$) of both Symbiodiniaceae species with their native microbiome alongside the Ab+Tx treated, EBR and bacteria co-cultures to provide baseline information on Symbiodiniaceae culture health and photophysiological performance.

## Symbiodiniaceae and bacterial cell density analysis
Cell counts were conducted through flow cytometry for each Symbiodiniaceae culture immediately before metabolomics samples were collected, immediately before and after co-culture mixing, and at the end of the growth period. Specifically, an aliquot of 100 μL was collected from each 1 mL sample, diluted 1:10 and directly used for flow cytometry analysis (CytoFLEX S, Beckman Coulter, CA, United States). Symbiodiniaceae cells were identified according to their chlorophyll a fluorescence (650 nm) and subsequently enumerated. Sample blanks ($n = 4$) were run alongside, and their average number of events subtracted from each sample (blank correction). Symbiodiniaceae flow cytometer gating strategy is shown in Figure S3a.

Total prokaryotic abundances were quantified on the same instrument by staining the cells with SYBR Green (1:10,000 final dilution). Bacteria cell counts were conducted on untreated and Ab+Tx treated cultures, EBR cultures immediately before co-culture mixing, and untreated, EBR and co-cultures at the end of the growth period. For each sample of each co-culture, $3 \times 200$ μL aliquots were taken at the time of sampling and fixed in glutaraldehyde (Sigma-Aldrich; 2% final concentration). The samples were analysed at a flow rate of 10 μL min$^{-1}$, with bacterial cells discriminated according to forward scatter (FSC), side scatter (SSC), and green fluorescence (SYBR Green, 488 nm). Bacteria flow cytometer gating strategy is shown in Figure S3b. To account for non-target staining and counts, sample blanks (sterile media only; $n = 5$) were stained and run alongside, and the average number of events from the blanks subtracted from each sample count (blank correction; Fig S3c).

### Symbiodiniaceae growth
For each Symbiodiniaceae species and co-culture mix (including bacteria-free and untreated subcultures), $2 \times 400$ μL aliquots were placed into a 48 well plate ($n = 2$ wells per culture). Plates were sealed with electrical tape and placed in an incubator at 26 °C with an irradiance of $70 \pm 5$ μmol photons m$^{-2}$ s$^{-1}$. Growth rates were estimated by measuring in vivo chlorophyll a fluorescence (relative fluorescence units) in a Tecan Spark plate reader[52] and measured as follows; 16 reads per well at excitation wavelength of 455 nm, emission wavelength of 630, 664, and 750 nm, gain at 80 nm; with 30 flashes at a frequency of 400 Hz; integration time of 20 μs; lag time of 0 μs, and settle time of 10 ms. Measurements were taken every 1-3 days for a total of 14 days and plates replaced in incubators immediately after each measurement (~30 s per plate). Specific growth rates ($\mu$) were calculated from the linear regression of the natural log of the in vivo fluorescence versus time during the exponential growth phase of cultures. Standard error of $\mu$ was calculated from $\mu$ values from biological replicates ($n = 5$) over the exponential growth period. Percentage growth enhancement was calculated as the difference between $\mu_{\text{co-culture}}$ and $\mu_{\text{EBR}}$ divided by $\mu_{\text{co-culture}}$.

### Photophysiology
Photophysiological performance of Symbiodiniaceae were assessed using a Soliense LIFT (Light Induced Fluorescence Transient)-FRR (Fast Repetition Rate) fluorometer (LIFT-FRRf; Soliense Inc. USA)[53]. All

cultures were low light (ca. 5-10 μmol photons m$^{-2}$ s$^{-1}$) acclimated for at least 30 min prior to measurements. An aliquot of 100 μL for each culture replicate (total $n = 30$, consisting of 5 replicates for each Symbiodiniaceae species [S. microadriaticum and B. minutum] for both EBR and untreated treatments) was transferred to the LIFT-FRRf optical chamber and diluted with 900 μL ASW. Excitation was delivered using a blue LED excitation source (peak excitation 470 nm), delivering single turnover fluorescent transients of 100 flashlets of 1.6 μs at 2.5 μs, followed by 127 flashlets of 1.6 μs, using a rapid Fluorescence Light Curve protocol (0, 10, 25, 50, 100, 250, 500, 750, 1000, 1250 μmol photons m$^{-2}$ s$^{-1}$), with each light step lasting 25 s and a collection of 2 acquisitions. All fluorescence yields were adjusted for baseline fluorescence using ASW + F/2. Light response protocols were used to derive the maximum PSII photochemical efficiency ($F_v/F_m$, dimensionless) and saturating light intensity ($E_K$, μmol photons m$^{-2}$ s$^{-1}$) using a model describing the light dependency of PSII photochemistry (as per Suggett et al. 2022). Values for the maximum light use efficiency ($\alpha$ [mol photon]$^{-1}$) and maximum relative rate of electron transport (rETR μmol electrons m$^{-2}$ s$^{-1}$) were also determined using a model describing the light dependency of electron transport (where ETR is the product of effective photochemical efficiency and light intensity for each light step (as per Hennige et al. 2008). Collectively we used $F_v/F_m$, $E_K$, $\alpha$ and rETR as proxies for tracking cellular health.

### Sampling for bacterial and metabolite composition
Seven days after the antibiotic treatment process, $3 \times 1$ mL aliquots were collected from each culture flask for flow cytometry cell counting of both bacteria and Symbiodiniaceae cells, bacteria community analysis via 16 S rRNA amplicon sequencing analysis, and Symbiodiniaceae photophysiological measurements (methods described in more detail below). The remaining Symbiodiniaceae cells were concentrated by centrifugation at $700 \times g$ for 10 mins at 26 °C, the media discarded, and Symbiodiniaceae pellets snap frozen in liquid nitrogen for metabolite profiling.

### Microbial community identification and sequences analysis
Bacterial DNA was extracted as per[54] using Qiagen Blood and Tissue DNA Extraction Kit (Qiagen, Germany) according to the manufacturer's instructions and the V3-V4 region of the 16 S rRNA gene was amplified using the 341 F (5′-TCGTCGGCAGCGTCAGATGTGTATAAGA GACAGCCTACGGGNGGCWGCAG-'3) and 805 R (5′-GTCTCGTGGGC TCGGAGATGTGTATAAGAGACAGGACTACHVGGGTATCTAATC-'3) primers[55] with Illumina adaptors underlined. PCR reactions consisted of 1 μL of template DNA, 5 μL of Velocity 5x Hi-Fi reaction buffer, 0.2 μL of 100 mM dNTP mix, 0.5 μL of high-fidelity velocity polymerase (Bioline, UK),1 μL of each 10 μM primer and water to the final volume of 25 μL. The PCR cycling conditions involved an initial denaturation step at 98 °C for 2 min, then 30 cycles at 98 °C for 30 s, 55 °C for 30 s and 72 °C for 30 s, followed by a final extension at 72 °C for 5 min. A positive (obtained in-house from a prior successful Symbiodiniaceae-associated microbiome analysis[18]) and negative (autoclaved 0.22 μm filtered media) control samples were included in the DNA extraction and velocity PCR amplification steps to verify successful extraction and control for contamination. The negative control did not produce a visible band following electrophoresis. Sample amplicons (excluding the positive and negative controls) were sequenced using the Illumina MiSeq platform ($2 \times 300$ bp) at the Australian Genome Research Facility (Melbourne, Australia). Raw data files in FASTQ format were deposited in the NCBI Sequence Read Archive (SRA) under PRJNA922609.

Raw FASTQ format files obtained from the 16 S rRNA gene amplicon sequencing were processed using the Quantitative Insights into Microbial Ecology (QIIME2) pipeline[56]. The DADA2 plugin (version 2019.1.0) was subsequently applied to remove chimeras, denoise and trim paired-end sequences[57]. Sequences were identified at the single

nucleotide threshold to produce amplicon sequence variants (ASVs). ASVs with reads below 0.01% in relative abundance, unassigned reads and ASVs corresponding to chloroplast or mitochondria were removed[58]. Rarefaction curves were produced to determine differences in sequencing depth between samples, with data then rarefied to 7800 reads per sample. Taxonomy was assigned using *classify-sklearn*[59] against the SILVA v138 database.

To visualise differences in bacterial community composition among locations and treatments, nonmetric multidimensional scaling ordinations (nMDS) were carried out using Bray-Curtis dissimilarity matrices in PAST (PAST v4.03). Differences in community structure were analysed using permutational multivariate analysis of variance (PERMANOVA) in PRIMER (v7) on log-10 transformed ASV count tables. A two-way PERMANOVA (Bray Curtis) of bacterial composition was conducted comparing Symbiodiniaceae species, treatments, and their interaction, followed by a one-way PERMANOVA with 9999 permutations to test for differences between untreated and Ab+Tx treated cultures of each Symbiodiniaceae species. To identify differentially abundant bacterial taxa between treatments (untreated and Ab+Tx treated), we used ANCOM-BC at the ASV and genus levels using the 'phyloseq', 'microbiome' and 'ANCOMBC' packages in R (v 4.3.0), with a holm adjusted *p* value. A Mann Whitney U test was then performed using M²IA[60] to analyse microbial composition changes at the genus level for both untreated and Ab+Tx treated cultures.

## Metabolite extraction

All subsequent steps were performed at 4 °C to prevent metabolite losses during extraction. Metabolite extractions, analysis and data processing are based on the methods described in[61]. To first remove residual salts (which affect GC-MS analysis), each pellet was resuspended in 500 µL cold (4 °C) MilliQ, gently agitated for 10 s, and centrifuged at 1,500 × *g* for 5 mins at 4 °C and the supernatant discarded. Pellets were frozen at -80 °C for 1 h and lyophilized at -105 °C for 18 h. The semi-polar metabolites were extracted by adding approx. 10 mg acid-washed glass beads to each pellet and 200 µL 100% cold (-20 °C) methanol spiked with 20 µg/mL final concentration of the internal standard D-sorbitol-6-¹³C, and cells were lysed using a bead mill at 50 Hz for 3 mins. A further 800 µL of 100% cold methanol (+ IS) was added to each cell slurry, and samples vortexed for exactly 1 min each. Cell debris was pelleted at 3,000 × *g* for 30 mins at 4 °C, and the supernatant transferred to a new 2 mL Eppendorf. To each cell debris, a further 1 mL 50% cold (−20 °C) methanol was added and samples vortexed for 30 s. Cell debris was pelleted at 3,000 × *g* for 30 mins at 4 °C, and the supernatant combined with the 100% methanol extracts. Samples were centrifuged at 16,000 × *g* for 15 mins at 4 °C, and 5 × 50 µL (250 µL total volume) dried in a glass insert in a concentrator at 30 °C.

## Online derivatisation and gas chromatography-mass spectrometry analysis

Dried samples for targeted analysis were derivatised online using the Shimadzu AOC6000 autosampler robot. Derivatisation was achieved by adding 25 µL of Methoxyamine Hydrochloride (30 mg/mL in Pyridine) followed by shaking at 37 °C for 2 h. Samples were then derivatised with 25 µL of *N,O-bis* (Trimethylsilyl)trifluoroacetamide with Trimethylchlorosilane (BSTFA with 1% TMCS, Thermo Scientific) for 1 h at 37 °C. The sample was left for 1 h before 1 µL was injected onto the GC column using a hot needle technique. Split (1:10) injections were done for each sample. The GC-MS system used comprised of an AOC6000 autosampler, a 2030 Shimadzu gas chromatograph and a TQ8040 quadrupole mass spectrometer (Shimadzu, Japan). The mass spectrometer was tuned according to the manufacturer's recommendations using tris-(perfluorobutyl)-amine (CF43). GC-MS was performed on a 30 m Agilent DB-5 column with 1 µm film thickness and 0.25 mm internal diameter column. The injection temperature (Inlet)

was set at 280 °C, the MS transfer line at 280 °C and the ion source adjusted to 200 °C. Helium was used as the carrier gas at a flow rate of 1 mL/min and Argon gas was used as the collision cell gas to generate the MRM product ion. The analysis of TMS samples was performed under the following temperature program; start at injection 100 °C, a hold for 4 minutes, followed by a 10 °C min⁻¹ oven temperature ramp to 320 °C following final hold off for 11 mins. Approximately 520 quantifying MRM targets were collected using Shimadzu Smart Database along with qualifier for each target which covers about 350 endogenous metabolites and multiple ¹³C labelled internal standards. Both chromatograms and MRMs were evaluated using the Shimadzu GCMS browser and LabSolutions Insight software.

## Metabolite data analysis

Metabolite data were normalised to peak area of the internal standard D-sorbitol-6-¹³C and then to cell debris protein content (as above). Metabolites with a percent relative standard deviation (RSD = standard deviation / mean) >30 % across all samples were removed from further analysis. To test for overall differences in metabolite pools between axenic and untreated cultures, statistical analysis were performed using MetaboAnalyst 4.0[62], where data were tested for normality and homogeneity, and cube root-transformed where necessary. Data were then evaluated by Principal Component Analysis (PCA) and supervised Partial Least Squares-Discriminant Analysis (PLS-DA). Univariate (T-tests) were performed to identify individual metabolites that varied significantly between the treatment groups. Significant individual metabolites were determined based on a False Discovery Rate (FDR) corrected significance value ($p_{adj} < 0.05$).

## Correlation analysis

Bacterial relative abundance data at the level of genus and metabolite relative abundance data was used to construct a correlation matrix of the microbe-metabolite data. A multivariate regression analysis of the microbe-metabolite data was performed using M²IA[60] and visualized as PLS-DA and tested for significance. The Spearman's correlation coefficient (*r*) of this regression was performed using M²IA[60]. Positive correlations and interactions with a significant *r* value and over 0.7 were then used to build a network in the Cytoscape software[63].

## Symbiodiniaceae-associated bacteria isolation

Bacteria were isolated by plating 10 µL of untreated *S. microadriaticum* and *B. minutum* cultures on 100% Marine Agar (n = 2 per species, BD Difco), incubated at 26 °C for 48 h and 10 random colonies picked from each Symbiodiniaceae species (5 from each plate), individually inoculated to sterile Marine Broth (BD Difco) and incubated at 26 °C for 12 h at 180 rpm. This process was repeated until purity. From each inoculation, DNA was extracted as described above and 16 S rRNA amplification performed using the primers 27 F (5'-AGAGTTT-GATCCTGGCTCAG-3') and 1492 R (5'-GGTTACCTTGTTACGACTT-3')[64]. Aliquots of each isolate were stored in glycerol at -80 °C. From these isolated strains, *Labrenzia alexandrii*, *Marinobacter* sp. and *Muricauda aquimarina* with the highest % match (>98%, E values = 0) were selected for further experiments (Supplementary Data 12).

## NanoSIMS preparation

To assess net nutrient uptake by both bacteria and Symbiodiniaceae cells in co-culture, EBR *S. microadriaticum* and *B. minutum* cultures at -50,000 cells mL⁻¹ were incubated in 10 mL 0.2 g·L − 1 NaH¹³CO³-ASW (99% ¹³C) (0.22 µm filter sterilised) + F/2 for 36 hours. Bacteria isolates (*L. alexandrii*, *Marinobacter* sp. and *M. aquimarina*) were plated as above, and after 3 days a colony of each was resuspended in 50 mL + ¹⁵NH₄Cl-media (0.010 g L⁻¹, 99% ¹⁵N, 0.22 µm filter sterilised), adjusted to pH 7 with 1 M KOH. Bacteria were incubated at 26 °C ± 2 °C with stirring at -120 rm for 14 hours (mid-exponential growth). Symbiodiniaceae and bacteria cells were counted via flow cytometry (as above),

centrifuged gently and resuspended in 10 mL unlabelled ASW + F/2 at $10^4$ and $10^6$ cells mL$^{-1}$ respectively, and each co-culture ($n = 3$ replicates of each co-culture) were set-up at a ratio of 10:1 Symbiodiniaceae:bacteria. Co-cultures were incubated at 26 °C for 6 hours, after which the 3 replicates were combined, pelleted via centrifugation and rinsed twice in 5 mL ASW. Cells pellets were fixed in 4% paraformaldehyde in ASW for 1 hr at 4 °C. Cells were centrifuged at 3,000 × g for 5 min, the supernatant was removed, and the pellet resuspended in 20 μL of ultrapure water and immediately dried at 35 °C on silicon wafers. The wafers were then stored in a desiccator until further processing.

Silicon wafers with attached sample sections were gold-coated and imaged with the NanoSIMS 50 L ion probe at the Centre for Microscopy, Characterisation and Analysis at the University of Western Australia. Surfaces of samples were bombarded with a 16-keV primary Cs$^+$ beam focused to a spot size of about 100 nm with a current of *ca.* 2 pA. Secondary molecular ions $^{12}C^{12}C^-$, $^{12}C^{13}C^-$, $^{12}C^{14}N^-$, and $^{12}C^{15}N^-$ were simultaneously collected in electron multipliers at a mass resolution (M/ΔM) of about 8,000. Charge compensation was not necessary. At least 20 images of different areas of the wafer (30-μm raster with 256 × 256 pixels) were recorded for each co-culture for all targeted secondary molecular ions by rastering the primary beam across the sample with a dwell time of 2 ms per pixel; 30 planes were recorded for each area. Image processing was performed using the ImageJ plugin OpenMIMS (National Resource for Imaging Mass Spectrometry, https://github.com/BWHCNI/OpenMIMS/wiki). After drift correction, the individual planes were summed and the $^{13}C/^{12}C$ or $^{15}N/^{14}N$ maps were expressed as a hue-saturation-intensity image, where the colour scale represents the Atom %. Assimilation of the isotope labels (atom % enrichment compared to unlabelled controls) was quantified for a minimum of 14 symbiont cells ($^{15}N$-labeled) and 40 bacteria ($^{13}C$-labeled) cells per co-culture by drawing individual regions of interest (ROIs) based on the silhouette of the $^{12}C^{14}N^-$ symbiont or bacteria cells using the ImageJ FIJI (v1.53c) plugin OpenMIMS (Fig. S5). The measured isotope ratios were converted to $^{15}N$ or $^{13}C$ atom fraction (Atom%) as per[65], which gives the percentage of a specific atom within the total number of atoms, and compared using one-way ANOVA with Tukey Post Hoc analysis, against the baseline from natural isotopic abundances calculated from the Atom % of $^{13}C$ or $^{15}N$ isotopes of unlabelled samples (21 unlabelled Symbiodiniaceae and 73 unlabelled bacteria).

## Verification of Indole-3-acetic acid (IAA) production by bacteria

The IAA production potential of the *L. alexandrii*, and *Marinobacter* sp. isolates were determined as follows; bacteria isolates were plated as above, and a single colony inoculated to sterilised Marine Broth and incubated at 26 °C for 12 h. Flow cytometry was used (as above) to determine the number of cells, and $n = 4$ flasks with 50 mL sterilised Marine Broth was inoculated with 10,000,000 cells/ml bacteria, and incubated at room temperature at 200 rpm. Bacterial growth was measured using OD on a spectrophotometer every 3 h for 12 hours, and growth curves were constructed to calculate late exponential phase. Isolates were then regrown as above, and $n = 4$ flasks with 50 mL sterilised Marine Broth supplemented with tryptophan (final concentration 50 nM) were each inoculated with 10,000,000 cells/ml and incubated at room temperature at 200 rpm until exponential growth phase was reached. The final tryptophan concentration of 50 nM was based on solid phase extraction (SPE) and LC-MS analysis (see *Extracellular IAA and Tryptophan Extractions* and *IAA and tryptophan quantification using LC-MS/MS* below) of EBR and untreated Symbiodiniaceae, which revealed an average exogenous concentration of 53 nM (±0.004, $n = 2$). After incubation, each bacteria/tryptophan culture broth was centrifuged at 2,000 × g for 30 min. The supernatant was collected and aliquoted into clean 50 ml conical centrifuge Falcon tubes. Both samples were flash frozen in liquid nitrogen and stored at −80 °C until IAA extraction.

## Intracellular IAA extractions

To quantify the IAA concentrations within the cell pellets, 2 ml of a 14:4:1 solution of $CH_3OH$, ultrapure $H_2O$, and $CH_2O_2$ was added, and tubes were sonicated for 2 minutes to lyse the cells. The cellular debris were subsequently centrifuged down, and quadruplicate samples of supernatant extract were freeze-dried at −80 °C. The sample residues were then reconstituted into 1 ml of 1.0 M $CH_2O_2$, and further purified on Evolute express CX columns (Biotage). A sixteen valve SUPELCO manifold was first cleaned with 99% methanol and MilliQ water. The columns were then attached and conditioned with $CH_3OH$ (99%) and 1.0 M $CH_2O_2$, before the reconstituted samples were loaded (note: the columns were never run dry). The samples were first eluted with 3 ml of 99% methanol, then 3 ml of 0.35 M $NH_4OH$ and finally 3 ml of 0.35 M in $NH_4OH$ 60% $CH_3OH$. All samples were then freeze dried at -80°C and reconstituted with 99% ultrapure high-pressure liquid chromatography (UHPLC) grade methanol.

### *Extracellular IAA and Tryptophan Extractions*

To quantify the IAA concentrations exuded by the bacterial cells, the supernatant of the bacteria cultures was first filtered through a 0.22 μm filter to remove all bacteria from the sample. Similarly, to quantify the tryptophan concentrations exuded by Symbiodiniaceae cells, the supernatant of EBR and untreated Symbiodiniaceae cultures were first filtered through a 0.22 μm filter to remove all cells from the sample. The absence of Symbiodiniaceae and/or bacteria was confirmed via flow cytometry (as above). The SUPELCO manifold and 6 cc 200 mg Oasis HBL cartridge filter columns were used to extract IAA and tryptophan from the samples. The columns were first conditioned with of ultrapure 20 ml $H_2O$, then 20 ml 99% $CH_3OH$, the bacterial supernatant was then loaded into the column 10 ml at a time. The samples were then eluted with 3 ml of 20% methanol, then 3 ml of 70% methanol. All samples were then freeze dried at −80 °C and reconstituted with 99% ultrapure high-pressure liquid chromatography (UHPLC) grade methanol.

## IAA and tryptophan quantification using LC-MS/MS

Known standards of IAA and tryptophan were used to calibrate the LC-MS/MS to mobile phase, run time and energy requirements for phytohormone fragmentation. A standard curve with the concentrations 0.5 μg/ml, 0.25 μg/ml, 0.125 μg/ml, 0.0625 μg/ml, 0.0312 μg/ml, 0.0156 μg/ml, 0.00781 μg/ml, 0.00390 μg/ml, and 0.000195 μg/ml was prepared.

IAA and tryptophan quantification was performed using a Shimadzu LCMS-8060 (Shimadzu, Kyoto, Japan) instrument containing a dual ion source (DUIS) interfaced to the Shimadzu Nexera X2 liquid chromatography system. All separation steps were performed on an Acquity UPLC HSS T3 column (1.8 μm, 2.1 × 150 mm) using a binary gradient of milliQ water (A) and methanol (B) for 15 minutes at a flow rate of 0.18 mL/min. The linear gradient program was run at: 0–10 min, 20–95% B; 10–11 min, 95-20% B; 11-15 min, 20% B. Column temperature was maintained at 30 °C. Samples were analysed in multiple reaction monitoring mode. The following transitions were monitored with a fragmentor voltage of 160 V and collision energy -6 eV. IAA (m/z 174 → 130) and $d_5$- IAA (m/z 179 → 135). The resulting peaks were then quantified using the manual integration tool on the Shimadzu LabSolutions CS analytical data management software (v3.8, Shimadzu, Kyoto, Japan). A nine-point calibration curve plot and constructed using analytical standards was then used to quantify unknown IAA and tryptophan concentrations.

## Symbiodiniaceae-bacteria co-culture generation

Cultures of EBR Symbiodiniaceae ($n = 5$ per species) were counted via flow cytometry (as above), centrifuged gently (1000 × g for 10 mins) and resuspended in ASW + F/2 at $10^4$ cells mL$^{-1}$. For IAA inoculations, 50 nM indole-3 acetic acid[5] was added to each culture. For bacteria co-

culture inoculations, bacteria isolates (*L. alexandrii* and *Marinobacter* sp.) were plated as above, and after 3 days, a colony of each was resuspended in 50 mL marine broth media ($n = 5$ per bacteria species, autoclaved and 0.22 μm filtered). Bacteria were incubated at $26 \pm 2\,°C$ with stirring at ~120 rpm for 14 hours (mid-exponential growth). A 100 μL aliquot of each bacteria culture was diluted to $1:10^5$ and stained with SYBR green and counted *via* flow cytometry (as above), centrifuged gently and resuspended in ASW + F/2 at $10^6$ cells mL$^{-1}$. Symbiodiniaceae and bacteria cells were mixed, with each co-culture ($n = 5$ replicates of each co-culture) set-up at a ratio of 10:1 Symbiodiniaceae:bacteria. Bacteria in the co-cultures were immediately quantified *via* flow cytometry (as above). The specific growth rates of untreated and EBR controls, EBR + IAA, and bacteria co-cultures were estimated by measuring in vivo chlorophyll *a* fluorescence (relative fluorescence units) in a Tecan Spark plate reader[52] and calculated as described above. Bacteria in the co-cultures were quantified via flow cytometry (as described above) at the end of the growth period and normalised to Symbiodiniaceae cell density.

### Reporting summary

Further information on research design is available in the Nature Portfolio Reporting Summary linked to this article.

## Data availability

Bacterial sequences for the isolates of *Labrenzia*, *Muricauda* and *Marinobacter* used have been uploaded to NCBI BioProject PRJNA922609. All data supporting our findings have been provided in the Supplementary Data files. Source data are provided with this paper. Raw GCMS data are available at https://doi.org/10.21228/M8Z132. NanoSIMS images are available at https://doi.org/10.6084/m9.figshare.24312778. Source data are provided with this paper.

## Code availability

Scripts used are available at https://doi.org/10.5281/zenodo.8385031

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

## Acknowledgements

This work was supported by Human Frontier Science Programme Long-term Postdoctoral fellowship LT000625/2018-L (awarded to J.L.M.), Australian Research Council Discovery Project grants DP180100838 (awarded to J-B.R. and J.R.S) and DP180100074 (awarded to D.J.S.) and DP200100091 (awarded to D.J.S. and J.L.M.). J-B.R. was supported by an Australian Research Council Fellowship (FT210100100).

## Author contributions

J.L.M, D.J.S., J.R.S., and J-B.R conceived and planned the research and analysed the data; J.L.M. performed most of the experiments; J-B.R. provided help with the solid phase extractions and sample preparation for NanoSIMS; J.B. and P.G. performed the NanoSIMS analysis; A.K. and U.K. performed the LC-MS analysis; N.S. and M.D.M. provided bioinformatics assistance; N.M.L.R provided help with the growth studies; and J.L.M and J-B.R. wrote the manuscript, with contributions from all of the authors.

## Competing interests

We declare we have no competing interests.
