## [Peer Review File · Nature Communications]

Coral endosymbiont growth is enhanced by metabolic interactions with bacteriaREVIEWER COMMENTS

Reviewer #1 (Remarks to the Author):

The manuscript “Coral endosymbiont growth is enhanced by metabolic interactions with bacteria” the potential metabolic interactions between coral microalgal endosymbionts (Symbiodiniaceae) and bacteria. The authors present an elegant approach to show that specific bacterial compositions correlate with significant shifts in the cellular metabolism of free-living Symbiodiniaceae. Specifically, they correlate the abundance of monosaccharides and indole-3-acetic acid (IAA) with the presence of specific bacteria and also show that two bacterial genera are involved in reciprocal exchanges of carbon and nitrogen with Symbiodiniaceae, resulting in growth improvements. The survey is conducted through a nicely planned experiment design, which helps elucidate key questions regarding some of the beneficial potential of bacteria in promoting Symbiodiniaceae’s growth. Moreover, the interkingdom interaction described in this work can support the search for other interactions that can also promote algae health and its use as probiotics for the coral holobiont. I believe that this manuscript will be of great interest to Nature Communications readers and helps bridge some gaps in our understanding of the mechanism of algae-bacteria interactions – which should also be expanded to surveys using in-hospite algae, when and if possible. Overall, I have an excellent impression of the paper and only a few questions and some minor suggestions to share.

Abstract:

Introduction:

Line 22: I suggest you add something about the phycosphere effect, such as: “...and their associated bacteria 22, creating a selective and specific phycosphere effect (i.e., a physical interface that can select specific members of the microbiome) (Bell et al., 1972 – and perhaps another ref, e.g., Seymour et al.)”

Bell, W. & Mitchell, R. Chemotactic and Growth Responses of Marine Bacteria to Algal Extracellular Products. *Biol. Bull.* 143, 265–277 (1972).

Seymour, J. R., Amin, S. A., Raina, J.-B. & Stocker, R. Zooming in on the phycosphere: the

ecological interface for phytoplankton–bacteria relationships. *Nature Microbiology* 2, 1–12 (2017).

Results and Discussion:

Overall: This section is well written, and the interactions and investigations are nicely performed and presented, except for a few parts of the text.

Lines 87-90: This is unclear; I don't think S table 6 is a friendly way to present this data. I recommend that authors be outspoken here and give the stats (in the text) on how much minimized the community was and whether this was significant.

Lies 90-101: This is critical data that was only superficially discussed. There is a new field of research on coral probiotics that has been focusing on the holobiont (including benefits to the algae), where this data is very much aligned. It is also helpful to add whether the differences reported were significant (add stats in the text, when possible, or only stick to significant differences, when reporting the increase or decrease of the relative abundance of metabolites).

Lines 131-133 – Please add a reference here.

Line 138 – How did you confirm the absence of bacteria? Please cite the fig/table showing the lack of bacterial cells here, not the table with the protocol used.

Line 149 – please also add the p-value for ^{13}C enriched *L. alexandrii*.

Line 152 – same here for the enriched *M. adhaerens* and *M. aquimarina*, the same way authors present it to ^{15}N enrichment in Symbiodiniaceae.

Line 167 – please state “based on...data”, referring to the method used to evaluate Symbiodiniaceae growth.

Figure 2 is fantastic, and so is the data presented.

Figure 3 b – please add the stats supporting these enrichments' significance.

I suggest the authors should highlight the importance of future efforts also to explore their findings in hospite algae-bacteria surveys, in non-treated cultures, evaluating the exchange of these metabolites in “normal conditions” (i.e., when the originally established microbiome is in place) to reinforce its natural prevalence and importance to the algae. The work is a beautiful proof-of-concept of specific metabolic exchanges and may guide future experiments to expand our knowledge on the topic.

Methods:

Lines 242-256: Why not also count the number of cells using a Neubauer chamber?

Line 264: Why these concentrations and these specific atb?

Lines 270-273: Have you also tested it via 16S qPCR (as mentioned in line 300)?

Lines 272-273: Please add more details here or in lines 297-303 describing the cytometry protocol and the detection limit (cell size). It is confusing overall, as the following section (“Symbiodiniaceae and bacterial cell density analysis”) also mentions a protocol, but that seems specific for before and after samples were collected for metabolomics. By only reading the M&M section, it is pretty confusing to understand how authors defined whether antibiotics have produced axenic cultures or just minimized the number and diversity of bacterial cells (which would also be ok, but it needs to be clear to readers). Extracting and sequencing 16S rDNA could be tricky, as dead cells could be picked, but qPCR would have been helpful here. I see the data in S Table 6, but I am not sure this is a friendly way to present this data. I recommend that you clearly show this result (lines 87-90).

Line 300: Please add more information on the protocols used, e.g., which primers, conditions, etc.

The methods section does not clearly indicate whether authors have evaluated Symbiodiniaceae's health before vs. after the use of atb (I reckon this is what it is called “untreated subcultures” refers to by reading the rest of the text, but this could be more clearly explained in lines 243-244). I understand the use of atb can be directly or indirectly harmful, and baseline information on the Symbiodiniaceae's performance/health status before any treatment was applied (i.e., not treated with atb) is key and should be highlighted, especially for further comparisons, with other studies. Although the focus is on

bacterial-driven improvements compared to atb-treated controls without bacterial inoculation, the baseline information indicating the importance of the native microbiome is also interesting.

Reviewer #2 (Remarks to the Author):

In this article, Matthews et al. investigate the role of bacterial associates in the physiology of the coral endosymbiont, Symbiodiniaceae, in its free-living stage. Correlative analyses led them to test whether bacteria of the *Marinobacter* and *Labrenzia* genus could provide a phytohormone, IAA, to two Symbiodiniaceae species. In vitro testing confirmed that both bacteria can produce IAA, while co-culture experiments showed that Symbiodiniaceae and bacteria can exchange nitrogen and carbon, and that Symbiodiniaceae can internalize IAA. Inoculation of Symbiodiniaceae with either bacterial culture, or IAA directly, increased its growth rate. This led the authors to their main conclusion, which is that bacterial associates can provide IAA to Symbiodiniaceae hosts, thereby participating in their growth.

There is no doubt that this research is significant and novel. To my knowledge, this is only the second study properly investigating the functional potential of Symbiodiniaceae-associated bacteria. The field of Symbiodiniaceae-bacteria interactions is still in its infancy, and studies such as this one will undoubtedly propel the field forward and provide invaluable information to understand the nature of such interactions, and their potential impact on coral health and physiology.

The paper is overall quite good and nicely brings together multidisciplinary approaches, including bacterial community profiling, nanoSIMS, metabolomics, and co-culturing.

Nonetheless, I have some major concerns and a number of minor comments that need to be addressed for the paper to be publishable, as I believe the main conclusions are not quite supported by the provided data.

Major comments:

1. I am concerned by the lack of negative controls (both for DNA extractions and for PCRs), as it is well known that contaminants are everywhere, especially in kit reagents. If no negative control was included, it must be clearly stated in the methods, and a sentence

must be added in the results explaining the limitations of the analysis because of the lack of negative controls. For example, I see a lot of *Corynebacterium*, *Cutibacterium*, *Staphylococcus* in Table S3, a lot of which may actually be contaminants. None were picked up in your analyses, but it could have skewed relative abundances and influenced the results for other, real, bacterial associates, and this needs to be explicitly mentioned.

2. The so-called extracellular bacteria-removed cultures lack detailed characterization, including: (i) whether extracellular bacteria were truly removed (eg include flow cytometry data); (ii) metabarcoding profiling before and after treatment, to confirm the decrease of the bacteria of interest.

3. While there is good evidence for it, the authors' main conclusion that *Labrenzia*/*Marinobacter* provide IAA to *Symbiodiniaceae* isn't quite well supported in my opinion. Indeed, the authors did not conduct community profiling following inoculation with *Labrenzia*/*Marinobacter* to confirm that they stay in the culture and are not just used for heterotrophic feeding. This point should be discussed in the manuscript, as the bacteria could have been eaten, therefore boosting *Symbiodiniaceae* growth, independently of IAA production, and could explain why IAA concentrations are so low in *Symbiodiniaceae*/bacteria co-cultures (ie a mix of *Symbiodiniaceae* internalizing IAA produced and of *Symbiodiniaceae* feeding on IAA-producing bacteria). Since no control for heterotrophic feeding (eg dead bacteria, that would provide a food source but no IAA) was included, this is a very plausible possibility. This alternative should be clearly pointed out in the discussion.

Minor comments:

The figures do not do justice to the results, and I suggest significant changes below to improve visualization and reader-friendliness.

The methods for *Symbiodiniaceae* inoculation with bacteria and IAA for Figure 3 are missing. Overall organization of the Material and Methods part is confusing, eg IAA paragraphs before NanoSIMS paragraph, while it is the opposite in the results: please follow the order in which the methods are applied in the results. See below for more details.

If the format allows it, I wonder if Fig S3 (and/or potentially Fig S1) could be brought to the main text, seeing as it really brings together the rest of the paper.

l. 29: I would describe IAA as a phytohormone here, rather than at l. 34, otherwise it seems a bit random to mention it specifically, and the reader might not be aware of the implication of this metabolite.

l. 31: replace “table” with “stable”

l. 32-33: “resulting in significant growth enhancement of the microalga” I would move this part at the end of the previous sentence, as the nanoSIMS work was not correlated with any growth measurements, but the correlative metabolite/bacteria analysis was.

l. 33-34: I would tone down this statement, as direct transfer between Symbiodiniaceae and bacteria was not observed (perhaps “we identified IAA as a potential metabolite provided by...”)

l. 57-58: ref 18 and 19 are identical in the reference list, so there might be a duplicate needing to be fixed here.

l. 58: regarding the core microbiome, it could be worth adding the following ref, which also finds *Labrenzia* and *Marinobacter* as core members in heat-evolved and wild-type *Cladocodium* cultures (Buerger, P., Vanstone, R. T., Maire, J., and van Oppen, M. J. H. (2022). Long-Term Heat Selection of the Coral Endosymbiont *Cladocodium C1acro* (Symbiodiniaceae) Stabilizes Associated Bacterial Communities. *International Journal of Molecular Sciences* 23, 4913. doi: 10.3390/IJMS23094913.). The ref could also be mentioned l. 64 in a quick additional sentence, seeing as the paper showed that the bacterial communities in heat-evolved Symbiodiniaceae changed compared to wild-type cultures, and therefore may play a role in thermal tolerance.

l. 93: Table S10 is referred to before Tables S8-9, please renumber the tables so that they are called in numerical order.

l. 94: While it is clear that the paper focuses on the effect of bacterial communities, it should still be briefly mentioned here that the antibiotics themselves could negatively impact the host through targeting chloroplasts/mitochondria, and thus independently of their associated bacterial consortia.

l. 103-105: I would expand a bit on what was actually done here and clearly state that the goal was to pair metabolites and bacteria from the results in the previous paragraph + analyze pairs that significantly increased/decreased with antibiotic treatment. Additionally, the analysis is not specifically targeted at the ‘growth rate’ (those metabolite-bacteria pairings could be related to any physiological change, really), so I suggest removing this part

of the sentence and keep it broader (eg something along the lines of ‘Using correlative analyses, we aimed to identify candidate metabolite-bacteria pairing that differ between treated and untreated cultures’).

Figure 1: I find the figure a little hard to read at first glance. I would recommend adding short prompts to panels c-e to clearly identify what we are looking at right away (eg above c: ‘Lower in Treated’, and above d ‘Higher in Treated’, and next to c-d written vertically ‘*Symbiodinium microadriaticum*’, or something like that)

I. 105: I believe *S. microadriaticum* should refer to Fig 1A, C, D, while *B. minutum* should refer to Fig 1B, E, F. Please modify throughout the paragraph.

I. 129: Is there a particular reason why only three strains are presented in Table S12, but 26 are mentioned in the text? I understand only the bacteria used further in the study are highlighted, but it would still be useful to mention the others. Additionally, it is still not quite clear from the text why these four genera were chosen, so perhaps here is a good time to (i) include all the bacteria isolated from the Symbiodiniaceae cultures in Table S12, (ii) tie previous literature (eg Motone et al 2020/the text currently in I. 132-134) to the correlative analysis, and (iii) in light of both (i) and (ii), and the fact that the antibiotic treatment seems to affect their relative abundance, clearly state that the study will go further with these three bacterial genera of interest.

I. 130: Taxonomic considerations: *Labrenzia alexandrii* is not the correct name but a homotypic synonym, and should be replaced with *Roseibium alexandrii* (see <https://lpsn.dsmz.de/genus/roseibium>). In fact, when blasting the sequences provided in Table S12, the closest hit does belong to *Roseibium alexandrii* according to NCBI taxonomy as well. I believe the SILVA databases would still give *Labrenzia* rather than *Roseibium*, so it might not be feasible to change it from Figure 1 and any metabarcoding data, so you should include a quick explanation of this taxonomic discrepancy in this paragraph.

I. 136: is there any proof that you actually removed all extracellular bacteria (considering it wasn’t the case for the AB+Tx treatment, Table S2)? The treatment seems quite drastic, so I would assume they were indeed, but I suggest pondering the statement/renaming this EBR group if there was no visualization/molecular work actually confirming it. Please provide flow cytometry measurements for these cultures as well.

I. 146: I would suggest using ‘co-culture’ throughout the manuscript, as ‘co-incubation’ can lead to confusion with the incubation with radioactive isotope step. Also it is unclear

whether this was performed with untreated or EBR Symbiodiniaceae cultures, please specify it somewhere. From the next paragraph (l. 162 onward) it seems that this was done with untreated, but seeing as the EBR treatment was just described a few lines prior, I initially assumed that the work was performed with EBR cultures. If this was done with untreated cultures, I suggest switching the EBR explanations (l. 134-138) to the paragraph starting l. 162 to avoid any confusion.

Figure 2: Since the carbon exchanges are mentioned first in the text, I would rearrange the figure to have the carbon panels first, (as a and c) and then the nitrogen panels. In panel c, I wonder if the *Labrenzia-Bminutum* photo is really the best one, as it shows no enrichment at all in the Symbiodiniaceae cell? If possible, I would suggest showing a photo with ¹⁵N enrichment in a Symbiodiniaceae cell.

l. 150-152: it could be because the bacteria were isolated from this culture, and not *Breviolum*, and they are perhaps more specialists than *Labrenzia*?

Figure 3a-b: Presented like this, the results are not super convincing as in (a) all growth curves are all over each other, and in (b) the different starting concentrations give a skewed vision (eg at first glance it could look like EBR in growing better than Untreated). I suggest for each panel to add a small inset with the growth rates from Figure S10 + the associated stats, then it would be very clear very quickly where the changes are.

l. 181: which microalgae? maybe worth mentioning how close they are to Symbiodiniaceae

l. 182: again, which 'specific bacteria'? if Gammaproteobacteria are known to produce it, it would be worth mentioning.

l. 184: how relevant is 10ug/ml tryptophan compared to what happens in a Symbiodiniaceae culture?

l. 190: Table S15 is referred to here, but Tables S12-14 have not yet. Again, some renumbering might be needed.

l. 193 and l. 196: Fig 3D (rather than 3C)?

l. 194: as mentioned above, this drastic difference could be due to the Symbiodiniaceae cells feeding on the *Labrenzia*/*Marinobacter*, rather than internalizing IAA. Please discuss this point.

Figure 3c-d: in the y-axis legend, please specify that 'cell' is 'bacterial cell' (could be confused with Symbiodiniaceae cell in panel d). Again, at first glance the results here are not super striking, and it takes noticing the 10⁻¹¹ vs 10⁻⁷ in the y-axis legends to clearly

understand the drastically different levels between Bacterial cultures and co-cultures. I suggest either (i) bringing panel d to the same scale as panel c (10^{-7} in the y-axis legend, and the numbers on the y-axis would have an extra 10^{-4}), or (ii) fusing panels c and d with the same y-axis, and perhaps a y-axis break to be able to show the very low levels clearly.

I. 203: no need to redefine EBR here.

I. 204: how does 50nm IAA compare to the levels measured in the bacterial cultures?

I. 206-207: I would quickly discuss here the much less pronounced effect on *B. minutum* (compared to *S. microadriaticum*). In light of the initial correlative analysis showing that IAA did not diminish in the antibiotic-treated *B. minutum*, perhaps the bacteria providing IAA to this culture were not removed by the treatment? Or it relies on other growth hormones?

I. 208: 'specific bacterial partners', I would remove 'specific' because many other bacteria than the ones you studied could be providing it.

I. 269-270: a sodium hypochlorite wash was mentioned in the results section for the EBR treatment, but not here. Please add it.

I. 281: no E culture is shown in this manuscript

I. 300: please replace '16S qPCR' by '16SrRNA gene metabarcoding'

I. 310: it is unclear to me how 'relative cell chlorophyll fluorescence' would give Symbiodiniaceae concentration in a flow cytometry analysis? Didn't the flow cytometer just count the cells anyway? This needs to be more detailed, ideally with a figure explaining the gating strategies.

I. 316-318: please provide a figure explaining the gating strategies to quantify bacteria.

I. 423: replace '16S identification' by 'taxonomic identification by 16S rRNA gene sequencing'. Additionally, I am not sure 'as described above' is accurate, as the sequences provided in Table S12 are ~1300-1400bp (ie full-length 16S rRNA gene), whereas the primers provided I. 324-326 would only amplify ~400bp. The part I. 420-424 is quite unclear: (i) it starts with *Labrenzia* as a goal, but then moves to *Labrenzia/Marinobacteria/Muricauda*; (ii) '10 random colonies picked' how did you end up with 26 colonies then (I. 129)? I would suggest writing its own paragraph for the bacterial isolation (without *Labrenzia* as a goal), clearly explaining how you isolated them and how you identified them (with the right primers).

I. 427: which nutrient broth was used?

I. 433: Tryptophan 50 nM was used in the figures (rather than 10ug/mL), please be

consistent.

Reviewer #3 (Remarks to the Author):

This study looks at the dependency of Symbiodiniacea on their associated bacteria by altering the bacterial composition of two coral endosymbiotic algae species during their free-living stage using detergent and antibiotic treatments. Results show a significant shift in algal cellular metabolism. Specifically, the abundance of monosaccharides and indole-3-acetic acid (IAA) were correlated with the presence of specific bacteria, including members of the *Labrenzia* and *Marinobacter* genera.

Single-cell stable isotope tracking revealed that these two bacterial genera are involved in reciprocal exchanges of carbon and nitrogen with Symbiodiniaceae, resulting in significant growth enhancement of the microalga. This work demonstrates how specific bacterial associates influence Symbiodiniaceae fitness.

This is a well written manuscript representing novel data on bacterial/algal associations that fills a significant gap in the literature associated with coral microbiology.

Methods are detailed and appear more than adequate. I have no suggestions.

Congratulations to the authors for this contribution

REVIEWER COMMENTS

Reviewer #1 (Remarks to the Author):

The manuscript “Coral endosymbiont growth is enhanced by metabolic interactions with bacteria” the potential metabolic interactions between coral microalgal endosymbionts (Symbiodiniaceae) and bacteria. The authors present an elegant approach to show that specific bacterial compositions correlate with significant shifts in the cellular metabolism of free-living Symbiodiniaceae. Specifically, they correlate the abundance of monosaccharides and indole-3-acetic acid (IAA) with the presence of specific bacteria and also show that two bacterial genera are involved in reciprocal exchanges of carbon and nitrogen with Symbiodiniaceae, resulting in growth improvements. The survey is conducted through a nicely planned experiment design, which helps elucidate key questions regarding some of the beneficial potential of bacteria in promoting Symbiodiniaceae’s growth. Moreover, the interkingdom interaction described in this work can support the search for other interactions that can also promote algae health and its use as probiotics for the coral holobiont. I believe that this manuscript will be of great interest to Nature Communications readers and helps bridge some gaps in our understanding of the mechanism of algae-bacteria interactions – which should also be expanded to surveys using in-hospite algae, when and if possible. Overall, I have an excellent impression of the paper and only a few questions and some minor suggestions to share.

We would like to sincerely thank the Reviewer for their positive comments.

Introduction:

1.1. Line 62: I suggest you add something about the phycosphere effect, such as: “...and their associated bacteria 22, creating a selective and specific phycosphere effect (i.e., a physical interface that can select specific members of the microbiome) (Bell et al., 1972 – and perhaps another ref, e.g., Seymour et al.)”

Bell, W. & Mitchell, R. Chemotactic and Growth Responses of Marine Bacteria to Algal Extracellular Products. *Biol. Bull.* 143, 265–277 (1972).

Seymour, J. R., Amin, S. A., Raina, J.-B. & Stocker, R. Zooming in on the phycosphere: the ecological interface for phytoplankton–bacteria relationships. *Nature Microbiology* 2, 1–12 (2017).

We thank the reviewer for the suggested additional text and supporting references. This has been added to the manuscript as follows:

L 60-62:

“[...] and their associated bacteria²², potentially creating a selective and specific phycosphere (i.e., the microenvironment enriched in photosynthates surrounding the microalgal cells^{3,23}).”

Results and Discussion:

1.2 Overall: This section is well written, and the interactions and investigations are nicely performed and presented, except for a few parts of the text.

Lines 87-90: This is unclear; I don't think S table 6 is a friendly way to present this data. I recommend that authors be outspoken here and give the stats (in the text) on how much minimized the community was and whether this was significant.

We thank the reviewer for this suggestion and have added the following text:

L 88 – 91:

*“Ab+Tx treatment significantly reduced the total number of bacteria cells per Symbiodiniaceae by 1.19 × in *S. microadriaticum* and 2.60 × in *B. minutum*, and altered the microbial profile of both Symbiodiniaceae species (PERMANOVA, $p < 0.001$; Tables S2&4).”*

We have also improved Table S6 by simplifying the columns, providing a colour scale to the fold change column to highlight the difference in the relative abundance of specific bacteria genera and ordered the rows according to fold change and Symbiodiniaceae species.

1.3 Lines 90-101: This is critical data that was only superficially discussed. There is a new field of research on coral probiotics that has been focusing on the holobiont (including benefits to the algae), where this data is very much aligned. It is also helpful to add whether the differences reported were significant (add stats in the text, when possible, or only stick to significant differences, when reporting the increase or decrease of the relative abundance of metabolites).

We thank the reviewer for their suggestion. We agree that our findings could provide new directions and intriguing avenues to explore in the field of coral probiotics. However, we felt it would be too speculative at this stage to imply how our findings here in Symbiodiniaceae culture metabolite profiles would relate to potential functional interactions offered by coral probiotics. In our summary, we have mentioned the importance of this study for coral probiotics on L 238:

“Our new evidence unlocking the metabolic connections between bacteria and Symbiodiniaceae partnerships will likely prove critical in ensuring the effectiveness of rapidly accelerating microbial-based reef intervention management tools⁴⁴, such as

probiotics designed to mitigate against stress-induced bleaching and disease^{45,46} or introduction of desired Symbiodiniaceae to host corals⁴⁷ that require optimised Symbiodiniaceae microbiomes.”

We have also reported the relevant statistics to support the significant differences observed for these compounds across treatments, as suggested (L 100-106):

“For *S. microadriaticum* and *B. minutum*, 7 and 28 metabolites significantly increased in relative abundance in response to Ab+Tx treatment respectively, including carbohydrates, myo-inositol, and precursors of the antioxidant glutathione (T-test, FDR < 0.05; Fig S3b&d, Table S10). Meanwhile, 3 and 6 metabolites significantly decreased in *S. microadriaticum* and *B. minutum*, respectively, including the growth promoting hormone indole-3-acetic acid (IAA), fatty acid metabolism by-products and TCA cycle intermediates (T-test, FDR < 0.05; Fig. S3b&d, Table S10).

1.4 Lines 131-133 – Please add a reference here.

As suggested, three references have been added to the text:

L 137:

“These bacteria are members of the core microbiome^{18,21} and have been implicated as playing functional roles for Symbiodiniaceae, including evidence of the translocation of antioxidant compounds¹⁶.”

16. Motone, K. et al. A zeaxanthin-producing bacterium isolated from the algal phycosphere protects coral endosymbionts from environmental stress. *Mbio* **11**, e01019-01019 (2020).

18. Lawson, C. A., Raina, J. B., Kahlke, T., Seymour, J. R. & Suggett, D. J. Defining the core microbiome of the symbiotic dinoflagellate, Symbiodinium. *Environmental microbiology reports* **10**, 7-11 (2018).

21. Maire, J. et al. Intracellular bacteria are common and taxonomically diverse in cultured and in hospite algal endosymbionts of coral reefs. *The ISME journal*, 1-15 (2021).

1.5 Line 138 – How did you confirm the absence of bacteria? Please cite the fig/table showing the lack of bacterial cells here, not the table with the protocol used.

We thank the reviewer for bringing this to our attention. We have now added Fig S2 to demonstrate the gating strategy used and provide an example of EBR verification and have added the flow cytometry data demonstrating Symbiodiniaceae and bacteria cell abundance after EBR treatment compared to untreated cells prior to co-culture mixing (Table S13). We have expanded on our methods for this analysis as follows:

L 281-300:

“Cell counts were conducted through flow cytometry for each Symbiodiniaceae culture immediately before metabolomics samples were collected, immediately before and after co-culture mixing, and at the end of the growth period. Specifically, an aliquot of 100 μ L was collected from each 1 mL sample, diluted 1:10 and directly used for flow cytometry analysis (CytoFLEX S, Beckman Coulter, CA, United States) to assess Symbiodiniaceae cell concentration by relative cell chlorophyll fluorescence (650 nm). Sample blanks (n = 4) were run alongside, and their average number of events subtracted from each sample (blank correction). Symbiodiniaceae flow cytometer gating strategy is shown in Figure S4a.

Total prokaryotic abundances were quantified on the same instrument by staining the cells with SYBR Green (1:10,000 final dilution). Bacteria cell counts were conducted on untreated and Ab+Tx treated cultures, EBR cultures immediately before co-culture mixing, and untreated, EBR and co-cultures at the end of the growth period. For each sample of each co-culture, 3 \times 200 μ L aliquots were taken at the time of sampling and fixed in glutaraldehyde (Sigma-Aldrich; 2% final concentration). The samples were analysed at a flow rate of 10 μ L min^{-1} , with bacterial cells discriminated according to forward scatter (FSC), side scatter (SSC), and green fluorescence (SYBR Green, 488 nm). Bacteria flow cytometer gating strategy is shown in Figure S4b. To account for non-target staining and counts, sample blanks (sterile media only; n = 5) were stained and run alongside, and the average number of events from the blanks subtracted from each sample count (blank correction; Fig. S4c).”

1.6 Line 149 – please also add the p-value for ^{13}C enriched *L. alexandrii*.

Line 152 – same here for the enriched *M. adhaerens* and *M. aquimarina*, the same way authors present it to ^{15}N enrichment in Symbiodiniaceae.

We have added the test and p values to the text as requested (L 152-159).

1.7 Line 167 – please state “based on...data”, referring to the method used to evaluate Symbiodiniaceae growth.

We have added a description of the method used and reference to the method as suggested.

L 171:

“Both bacteria enhanced the growth of each Symbiodiniaceae strain compared to the controls based on chlorophyll a fluorescence³⁹.”

1.8 Figure 2 is fantastic, and so is the data presented.

We thank the Reviewer for their compliment.

1.9 Figure 3 b – please add the stats supporting these enrichments' significance.

We thank the Reviewer for pointing out this error and have included the test (ANOVA) in the figure legend.

1.10 I suggest the authors should highlight the importance of future efforts also to explore their findings in hospite algae-bacteria surveys, in non-treated cultures, evaluating the exchange of these metabolites in “normal conditions” (i.e., when the originally established microbiome is in place) to reinforce its natural prevalence and importance to the algae. The work is a beautiful proof-of-concept of specific metabolic exchanges and may guide future experiments to expand our knowledge on the topic.

We thank the Reviewer for their suggestions and have added the following text:

L 236-238: “We are likely just scratching the surface of the chemical exchanges occurring between Symbiodiniaceae and bacteria, with many more important chemical currencies yet to be identified.”

Methods:

1.11 Lines 242-256: Why not also count the number of cells using a Neubauer chamber?

We used in vivo relative chlorophyll a fluorescent units to estimate cell density to avoid opening the plates for sub-sampling (and potentially contaminating the cultures with non-target bacteria). This sub-sampling would have been required for flow cytometry or haemocytometer counts.

1.12 Line 264: Why these concentrations and these specific atb?

We thank the reviewer for highlighting this point. The selection of antibiotics and concentrations were chosen to cover a range and antibiotic actions and were based on publications that successfully removed bacteria from phytoplankton cultures. To better reflect these decisions, we have added columns to Table S1 to detail the antibiotic classes, bactericidal effects and the references used to inform their selection and concentrations. These references have also been added to antibiotic treatment methods section of the main text (L 268).

In addition, while editing Table S1 and the main text, we noticed that Kanamycin ($50 \mu\text{g mL}^{-1}$) was missing from our list. This antibiotic has now been added to these sections.

1.13 Lines 270-273: Have you also tested it via 16S qPCR (as mentioned in line 300)?

We apologise for this typographic mistake, we meant 16S rRNA gene amplicon sequencing. This sentence has now been corrected and it reads:

L 351:

“Seven days after the antibiotic treatment process, 3 × 1 mL aliquots were collected from each culture flask for flow cytometry cell counting of both bacteria and Symbiodiniaceae cells, bacteria community analysis via 16S rRNA gene metabarcoding.”

1.14 Lines 272-273: Please add more details here or in lines 297-303 describing the cytometry protocol and the detection limit (cell size). It is confusing overall, as the following section (“Symbiodiniaceae and bacterial cell density analysis”) also mentions a protocol, but that seems specific for before and after samples were collected for metabolomics. By only reading the M&M section, it is pretty confusing to understand how authors defined whether antibiotics have produced axenic cultures or just minimized the number and diversity of bacterial cells (which would also be ok, but it needs to be clear to readers). Extracting and sequencing 16S rDNA could be tricky, as dead cells could be picked, but qPCR would have been helpful here. I see the data in S Table 6, but I am not sure this is a friendly way to present this data. I recommend that you clearly show this result (lines 87-90).

We thank the reviewer for bringing this to our attention. As described in response to comment 1.5 above, we have expanded our methods descriptions, and added our flow cytometry gating strategy (Fig. S2) and data for the verification of EBR cultures prior to co-culture mixing. As mentioned above, 16S data may have included the intercellular bacteria, and therefore misrepresent estimates of bacteria removal.

1.15 Line 300: Please add more information on the protocols used, e.g., which primers, conditions, etc.

The amplicon sequencing methods are described in more detail under the heading “Microbial community identification and sequences analysis”. However, the reviewer has spotted an error in our methods here as we initially wrote “16S qPCR” but meant “16S rRNA gene amplicon sequencing analysis”. We have amended the text as following:

L 351:

“[...] via 16S rRNA amplicon sequencing analysis, and Symbiodiniaceae photophysiological measurements (methods described in more detail below).”

1.16 The methods section does not clearly indicate whether authors have evaluated Symbiodiniaceae’s health before vs. after the use of atb (I reckon this is what it is called “untreated subcultures” refers to by reading the rest of the text, but this could be more clearly explained in lines 243-244). I understand the use of atb can be directly or indirectly harmful, and baseline information on the Symbiodiniaceae’s performance/health status before any treatment was applied (i.e., not treated with atb) is key and should be highlighted, especially for further comparisons, with other studies. Although the focus is on bacterial-driven improvements compared to atb-treated controls without bacterial inoculation, the baseline information indicating the

importance of the native microbiome is also interesting.

The reviewer is correct, we included untreated (no antibiotic or detergent treatments) controls and EBR controls, to provide baseline information on the performance/health of Symbiodiniaceae with their native microbiome and no microbiome, respectively. As mentioned above in response to comment 1.12, we selected the antibiotics and concentrations after carefully considering the existing literature. We verified that the antibiotic and detergent treatments we used did not induce photophysiological damage (Fig. S2, Table S7) and Symbiodiniaceae growth, while reduced, was not inhibited (Fig. 3 and Table S2). To clarify this point, we have added the following:

L 283:

“We maintained untreated control cultures (i.e., no detergent or antibiotic treatment; n = 5) of both Symbiodiniaceae species with their native microbiome alongside the Ab+Tx, EBR and bacteria co-cultures to provide baseline information on Symbiodiniaceae culture health and photophysiological performance.”

Reviewer #2 (Remarks to the Author):

In this article, Matthews et al. investigate the role of bacterial associates in the physiology of the coral endosymbiont, Symbiodiniaceae, in its free-living stage. Correlative analyses led them to test whether bacteria of the Marinobacter and Labrenzia genus could provide a phytohormone, IAA, to two Symbiodiniaceae species. In vitro testing confirmed that both bacteria can produce IAA, while co-culture experiments showed that Symbiodiniaceae and bacteria can exchange nitrogen and carbon, and that Symbiodiniaceae can internalize IAA. Inoculation of Symbiodiniaceae with either bacterial culture, or IAA directly, increased its growth rate. This led the authors to their main conclusion, which is that bacterial associates can provide IAA to Symbiodiniaceae hosts, thereby participating in their growth.

There is no doubt that this research is significant and novel. To my knowledge, this is only the second study properly investigating the functional potential of Symbiodiniaceae-associated bacteria. The field of Symbiodiniaceae-bacteria interactions is still in its infancy, and studies such as this one will undoubtedly propel the field forward and provide invaluable information to understand the nature of such interactions, and their potential impact on coral health and physiology.

The paper is overall quite good and nicely brings together multidisciplinary approaches, including bacterial community profiling, nanoSIMS, metabolomics, and co-culturing. Nonetheless, I have some major concerns and a number of minor comments that need to be addressed for the paper to be publishable, as I believe the main conclusions are not quite supported by the provided data.

We sincerely thank Reviewer 2 for appreciating the significance of our study and for their constructive comments.

Major comments:

2.1 I am concerned by the lack of negative controls (both for DNA extractions and for PCRs), as it is well known that contaminants are everywhere, especially in kit reagents. If no negative control was included, it must be clearly stated in the methods, and a sentence must be added in the results explaining the limitations of the analysis because of the lack of negative controls. For example, I see a lot of *Corynebacterium*, *Cutibacterium*, *Staphylococcus* in Table S3, a lot of which may actually be contaminants. None were picked up in your analyses, but it could have skewed relative abundances and influenced the results for other, real, bacterial associates, and this needs to be explicitly mentioned.

We thank the Reviewer for highlighting this point. We first apologise for the error in our description of the methods, as a negative control (autoclaved 0.22 µm filtered media) was indeed included in the DNA extraction and velocity PCR amplifications steps.

However, we decided not to submit the negative control for MiSeq sequencing, as it did not show a band in the gel. Nevertheless, we agree that sequencing the negative controls (if possible, given the detection threshold) would have supported our data.

We have added the following text to the main text to reflect this point:

L 369:

“A positive (obtained in-house from a prior successful Symbiodiniaceae associated microbiome analysis¹⁸) and negative (autoclaved 0.22 µm filtered media) control samples were included in the DNA extraction and velocity PCR amplification steps to verify successful extraction and control for contamination. The negative control did not produce a visible band following electrophoresis. Sample amplicons (excluding the positive and negative controls) were sequenced using the Illumina MiSeq platform (2 × 300 bp) at the Australian Genome Research Facility (Melbourne, Australia).”

2.2 The so-called extracellular bacteria-removed cultures lack detailed characterization, including: (i) whether extracellular bacteria were truly removed (eg include flow cytometry data); (ii) metabarcoding profiling before and after treatment, to confirm the decrease of the bacteria of interest.

We thank the reviewer for bringing this to our attention. As described in response to comments 1.5 above, we have expanded our methods descriptions, and added our flow cytometry gating strategy (Fig S2) and data for the verification of EBR cultures prior to co-culture mixing.

We have avoided the term axenic throughout our manuscript. The difficulty in using 16S amplicon sequencing to verify the absence of bacteria in Symbiodiniaceae is two-fold:

- 1. Antibiotic treatments will kill bacteria, but their DNA or cell debris may persist in the cultures, giving false positive.*
- 2. As previously reported (c.f. Maire et al. 2021), our methods might not remove intercellular bacteria, hence we termed the cultures “extracellular bacteria-removed”.*

We therefore decided to rely on bacterial growth assessment on agar plates and blank corrected flow cytometry counts, which would detect live, exogenous bacterial cells.

2.3 While there is good evidence for it, the authors’ main conclusion that Labrenzia/Marinobacter provide IAA to Symbiodiniaceae isn’t quite well supported in my opinion. Indeed, the authors did not conduct community profiling following inoculation with Labrenzia/Marinobacter to confirm that they stay in the culture and are not just used for heterotrophic feeding. This point should be discussed in the manuscript, as the bacteria could have been eaten, therefore boosting Symbiodiniaceae growth, independently of IAA production, and could explain why IAA concentrations are so low in Symbiodiniaceae/bacteria co-cultures (ie a mix of Symbiodiniaceae internalizing IAA produced and of Symbiodiniaceae feeding on IAA-producing bacteria). Since no control for heterotrophic feeding (eg dead bacteria, that would provide a food source but no IAA) was included, this is a very plausible possibility. This alternative should be clearly pointed out in the discussion.

We thank the reviewer for their insightful and important comment. Although we did not collect a sample for end-point community profiling, we have been able to confirm that Labrenzia and Marinobacter remained in culture by re-using samples originally collected during the co-culture experiments for Symbiodiniaceae counts (previously fixed with glutaraldehyde). We have now quantified the bacteria immediately after co-culture mixing (day 0), and 14 days post co-culture mixing, and the bacterial density per Symbiodiniaceae increased on average by 8.7 times for Labrenzia and by 12.2 times for Marinobacter. We have included this data (Table S15) which demonstrates that the bacterial population in the co-cultures not only remained, but substantially increased through the experiment. We have added a sentence in the discussion surrounding this data, which reads:

L 172:

“Both bacteria enhanced the growth of each Symbiodiniaceae strain compared to the controls based on chlorophyll a fluorescence (ANOVA; FDR < 0.01, Fig 4a, Table S8), and each bacterium remained in culture and substantially increased in abundance during the 14 days (8.7 times more Labrenzia and 12.2 more Marinobacter per Symbiodiniaceae cells; Table S15).”

It is also important to point out that nutrient transfers between the bacterial cells and the Symbiodiniaceae were not observed when the dinoflagellates were co-cultured with Muricauda (Figure 2b). This means that no significant exudation of nitrogen-rich compounds and no heterotrophic feeding on the bacteria occurred in that co-cultures.

While we recognise there is still much to be revealed in terms of Symbiodiniaceae-bacteria interactions, we feel confident the evidence we present indicates that Symbiodiniaceae uptake and use bacterial-secreted IAA for growth, and that heterotrophic feeding, while possible, was not solely responsible for the increased growth associated with these co-cultures.

We have added the following text:

L 222-224:

“While Symbiodiniaceae heterotrophic feeding on the bacteria¹⁴ could also provide a source of IAA, this study demonstrates the capacity for IAA secretion by associated bacteria, and the up-take of exogenous IAA which enhanced the growth of Symbiodiniaceae.”

Minor comments:

The figures do not do justice to the results, and I suggest significant changes below to improve visualization and reader-friendliness.

2.4 The methods for Symbiodiniaceae inoculation with bacteria and IAA for Figure 3 are missing. Overall organization of the Material and Methods part is confusing, eg IAA paragraphs before NanoSIMS paragraph, while it is the opposite in the results: please follow the order in which the methods are applied in the results. See below for more details.

We thank the reviewer for highlighting this error, and we have added the following text for the inoculation of Symbiodiniaceae with IAA and bacteria:

L 579-596:

“Symbiodiniaceae-bacteria co-culture generation

*Cultures of EBR Symbiodiniaceae (n = 5 per species) were counted via flow cytometry (as above), centrifuged gently (1000 × g for 10 mins) and resuspended in ASW + F/2 at 10⁴ cells mL⁻¹. For IAA inoculations, 50 nM indole-3 acetic acid⁵ was added to each culture. For bacteria co-culture inoculations, bacteria isolates (*L. alexandrii*, *Marinobacter sp.* and *M. aquimarina*) were plated as above, and after 3 days, a colony of each was resuspended in 50 mL marine broth media (n = 5 per bacteria species, autoclaved and 0.22 μm filter sterilised). Bacteria were incubated at 26 ± 2°C with stirring at ~120 rpm for 14 hours (mid-exponential growth). A 100 μL aliquot of each bacteria culture was diluted to 1:10⁵ and stained with SYBR green and counted via flow cytometry (as above), centrifuged gently and resuspended in ASW + F/2 at 10⁶ cells mL⁻¹. Symbiodiniaceae and bacteria cells were mixed, with each co-culture (n = 5 replicates of each co-culture) set-up at a ratio of 10:1 Symbiodiniaceae:bacteria. Symbiodiniaceae specific growth rates of untreated and EBR controls, EBR+IAA, and bacteria co-cultures were estimated by measuring in vivo*

chlorophyll a fluorescence (relative fluorescence units) in a Tecan Spark plate reader³⁹ and calculated as described above.”

We have also re-ordered the methods according to the order used in the results as suggested.

2.5 If the format allows it, I wonder if Fig S3 (and/or potentially Fig S1) could be brought to the main text, seeing as it really brings together the rest of the paper.

We thank the reviewer for this suggestion, and we have now brought Figure S3 into the main text (now Figure 1).

2.6 l. 29: I would describe IAA as a phytohormone here, rather than at l. 34, otherwise it seems a bit random to mention it specifically, and the reader might not be aware of the implication of this metabolite.

We have moved the text from L 34 to L29 as follows:

L 28-29:

“Indeed, the abundance of monosaccharides and the key phytohormone indole-3-acetic acid (IAA)...”

2.7 l. 31: replace “table” with “stable”

We thank the reviewer for pointing out this error and have correct the text to:

L 31:

“Single-cell stable isotope”

2.8 l. 32-33: “resulting in significant growth enhancement of the microalga” I would move this part at the end of the previous sentence, as the nanoSIMS work was not correlated with any growth measurements, but the correlative metabolite/bacteria analysis was.

We thank the reviewer for their suggestions. Instead of moving the end of this sentence at the end of the preceding one, we have edited the following sentence to incorporate this statement (which avoids a repetition). The edited sentence now reads:

L 33- 34:

“We identified the provision of IAA by Labrenzia and Marinobacter, and this metabolite caused a significant growth enhancement of Symbiodiniaceae.”

2.9 l. 33-34: I would tone down this statement, as direct transfer between Symbiodiniaceae and bacteria was not observed (perhaps “we identified IAA as a potential metabolite provided by...”)

We have adjusted the tone as suggested. The text now reads as follows:

L 32:

“[...] reciprocal exchanges of carbon and nitrogen with Symbiodiniaceae. We identified the provision of IAA by Labrenzia and Marinobacter, and this metabolite caused a significant growth enhancement of Symbiodiniaceae.”

2.10 l. 57-58: ref 18 and 19 are identical in the reference list, so there might be a duplicate needing to be fixed here.

We thank the reviewer for pointing out this error and have removed the duplicate.

2.11 l. 58: regarding the core microbiome, it could be worth adding the following ref, which also finds Labrenzia and Marinobacter as core members in heat-evolved and wild-type Cladocopium cultures (Buerger, P., Vanstone, R. T., Maire, J., and van Oppen, M. J. H. (2022). Long-Term Heat Selection of the Coral Endosymbiont Cladocopium C1acro (Symbiodiniaceae) Stabilizes Associated Bacterial Communities. International Journal of Molecular Sciences 23, 4913. doi: 10.3390/IJMS23094913.). The ref could also be mentioned l. 64 in a quick additional sentence, seeing as the paper showed that the bacterial communities in heat-evolved Symbiodiniaceae changed compared to wild-type cultures, and therefore may play a role in thermal tolerance.

We thank the reviewer for the reference suggestion and have included this reference in the two positions suggested (now Lines 57 and 65) and added the following text:

L 62:

“Finally, bacterial communities associated with heat-evolved strains of Cladocopium are more stable than wild-type strains²⁰, and specific bacteria can produce the potent antioxidant zeaxanthin¹⁶, [...]”

2.12 l. 93: Table S10 is referred to before Tables S8-9, please renumber the tables so that they are called in numerical order.

We thank the reviewer for pointing out this error and have renumbered the tables and references to them in the text.

2.13 l. 94: While it is clear that the paper focuses on the effect of bacterial communities, it should still be briefly mentioned here that the antibiotics themselves could negatively impact the host through targeting chloroplasts/mitochondria, and thus independently of their associated bacterial consortia.

We thank the reviewer for their comment. We have responded to a similar comment (1.12) by Reviewer 1. To summarise, we selected the antibiotics and concentrations to cover a range and antibiotic actions, based on publications that successfully removed bacteria from phytoplankton cultures. We have added columns to Table S1 to explain the antibiotic classes, bactericidal effects and the references used to inform their selection and concentrations. These references have also been added to antibiotic treatment methods section of the main text (L271). Kanamycin ($50 \mu\text{g mL}^{-1}$) was missing from our list but has now been added to these sections.

2.14 l. 103-105: I would expand a bit on what was actually done here and clearly state that the goal was to pair metabolites and bacteria from the results in the previous paragraph + analyze pairs that significantly increased/decreased with antibiotic treatment. Additionally, the analysis is not specifically targeted at the ‘growth rate’ (those metabolite-bacteria pairings could be related to any physiological change, really), so I suggest removing this part of the sentence and keep it broader (eg something along the lines of ‘Using correlative analyses, we aimed to identify candidate metabolite-bacteria pairing that differ between treated and untreated cultures’).

We agree with the reviewer and have edited the text as suggested:

L 108:

“Using correlative analysis, we aimed to identify candidate metabolites-bacteria combinations that differed between the untreated and treated Symbiodiniaceae cultures.”

2.15 Figure 1: I find the figure a little hard to read at first glance. I would recommend adding short prompts to panels c-e to clearly identify what we are looking at right away (eg above c: ‘Lower in Treated’, and above d ‘Higher in Treated’, and next to c-d written vertically ‘Symbiodinium microadriaticum’, or something like that)

We thank the reviewer and have edited the figure to improve clarity. We have moved all Symbiodinium panels on the left-hand side and all Breviolum panels on the left-hand side. We believe the revised figure is now easier to follow thanks to these suggestions.

2.16 l. 105: I believe S. microadriaticum should refer to Fig 1A, C, D, while B. minutum should refer to Fig 1B, E, F. Please modify throughout the paragraph.

We thank the reviewer for pointing out these errors and have renamed the references to the figures in the text.

2.17 l. 129: Is there a particular reason why only three strains are presented in Table S12, but 26 are mentioned in the text? I understand only the bacteria used further in the study are highlighted, but it would still be useful to mention the others. Additionally, it is still not quite clear from the text why these four genera were chosen, so perhaps here is a good time to (i) include all the bacteria isolated from the Symbiodiniaceae cultures in Table S12, (ii) tie previous literature (eg Motone et al 2020/the text currently in l. 132-134) to the correlative analysis, and (iii) in light of both (i) and (ii), and the fact that the antibiotic treatment seems to affect their relative abundance, clearly state that the study will go further with these three bacterial genera of interest.

We apologise for the confusion and have provided additional clarification about our bacteria isolations. We successfully isolated 26 bacterial strains but 16S analysis yielded low sequence quality for 9 of them. The remaining 17 strains matched closely to 7 different species (based on their 16S rRNA gene). We have now included sequences for all 17 isolates in Table S12, and provided clearer methods surrounding the isolation of these bacteria.

To better emphasize our choice of these three bacteria strains, we have modified and reorganised the paragraph to the following:

L 137:

“These bacteria are members of the Symbiodiniaceae core microbiome^{18,21} and may have important functions for these dinoflagellates, including the potential translocation of antioxidant compounds¹⁶ Given the ubiquitous presence of Labrenzia, Marinobacter, and Muricauda across Symbiodiniaceae genera¹⁸ as well as their strong correlation with multiple chemicals identified here, we quantified metabolic exchanges between members of these genera and Symbiodiniaceae using nanoscale secondary ion mass spectrometry (NanoSIMS).”

We have also refined our methods to clarify that we isolated bacteria from both strains of Symbiodiniaceae, and selected the isolates for co-culturing that had the highest percentage match in the NCBI database which was not originally reflected in our methods but has now been added. Text in the methods is now as follows:

L 461:

“Symbiodiniaceae-associated bacteria isolation

*Bacteria were isolated by plating 10 µL of untreated *S. microadriaticum* and *B. minutum* cultures on 100% Marine Agar (n = 2 per species, BD Difco), incubated at 26°C for 48 h and 10 random colonies picked from each Symbiodiniaceae species (5 from each plate), individually inoculated to sterilised Marine Broth (BD Difco) and incubated at 26°C for 12 h at 180 rpm. This process was repeated until purity. From each inoculation, was extracted as*

described above and 16S rRNA amplification performed using the primers 27F (5'-AGAGTTTGATCCTGGCTCAG-3') and 1492R (5'-GGTACCTTGTTACGACTT-3')⁶³. From these isolated strains, *Labrenzia* (*Roseibium*) *alexandrii*, *Marinobacter* sp. and *Muricauda aquimarina* with the highest % match (> 98%, E values = 0) were selected for further experiments (Table S12).”

2.18 l. 130: Taxonomic considerations: *Labrenzia alexandrii* is not the correct name but a homotypic synonym, and should be replaced with *Roseibium alexandrii* (see <https://lpsn.dsmz.de/genus/roseibium>). In fact, when blasting the sequences provided in Table S12, the closest hit does belong to *Roseibium alexandrii* according to NCBI taxonomy as well. I believe the SILVA databases would still give *Labrenzia* rather than *Roseibium*, so it might not be feasible to change it from Figure 1 and any metabarcoding data, so you should include a quick explanation of this taxonomic discrepancy in this paragraph.

The reviewer is correct, the SILVA database returns Labrenzia, not Roseibium. Labrenzia is the synonym used in most of the current literature on Symbiodiniaceae bacteria and was the taxonomic assignment returned in the 16S sequencing results. Unless the Reviewer and Editor strongly disagree, we would prefer to keep the synonym Labrenzia in our text and figures to importantly align with the literature and sequencing results. However, we have added the name Roseibium to the isolate 16S methods (as above) and results (Table S12). We have also included the synonym Roseibium to the abstract (Line 30), and the first mention of Labrenzia in the Results and Discussion (Line 93).

*We would also like to thank the Reviewer as upon re-examining the NCBI result of the isolate DNA, we have decided to adopt the broader taxonomy for the *Marinobacter* isolate (*Marinobacter* sp.) as the NCBI Blast search no longer returns *M. adhaerens* as the top match. We have edited the text throughout to reflect this change.*

2.19 l. 136: is there any proof that you actually removed all extracellular bacteria (considering it wasn't the case for the AB+Tx treatment, Table S2)? The treatment seems quite drastic, so I would assume they were indeed, but I suggest pondering the statement/renaming this EBR group if there was no visualization/molecular work actually confirming it. Please provide flow cytometry measurements for these cultures as well.

We thank the reviewer for further highlighting this gap. Please refer to our responses to comments 1.5 and 2.2 above regarding verification of EBR cultures via flow cytometry and plating, and reasoning for the absence of molecular work.

2.20 l. 146: I would suggest using ‘co-culture’ throughout the manuscript, as ‘co-incubation’ can lead to confusion with the incubation with radioactive isotope step.

We have amended the text as suggested (L 152 and 154).

2.21 Also it is unclear whether this was performed with untreated or EBR Symbiodiniaceae cultures, please specify it somewhere. From the next paragraph (l. 162 onward) it seems that this was done with untreated, but seeing as the EBR treatment was just described a few lines prior, I initially assumed that the work was performed with EBR cultures. If this was done with untreated cultures, I suggest switching the EBR explanations (l. 134-138) to the paragraph starting l. 162 to avoid any confusion.

We apologise for the confusion and have added reference to the use of the EBR cultures (L 149). In addition, we have rearranged the paragraph to better reflect these decisions:

L 139:

*“Given the ubiquitous presence of *Labrenzia*, *Marinobacter*, and *Muricauda* across Symbiodiniaceae genera¹⁸ as well as their strong correlation with multiple chemicals identified here, we quantified metabolic exchanges between members of these genera and Symbiodiniaceae using nanoscale secondary ion mass spectrometry (NanoSIMS). We completely removed extracellular bacteria from *S. microadriaticum* and *B. minutum* cultures via 4 rounds of filtering, rinsing with 6% sodium hypochlorite and 48 h of incubation with antibiotics (Table S1), resulting in cultures we henceforth termed “extracellular bacteria-removed” (EBR; Table S13), making them suitable for co-culture experiments with individual bacterial isolates. Prior to co-culturing, the three bacterial strains (*L. alexandrii*, *Marinobacter* sp., and *M. aquimarina*; Table S12) were pre-enriched with ¹⁵N, and the EBR cultures of the two Symbiodiniaceae species were pre-enriched with the rare stable isotope ¹³C”.*

2.22 Figure 2: Since the carbon exchanges are mentioned first in the text, I would rearrange the figure to have the carbon panels first, (as a and c) and then the nitrogen panels. In panel c, I wonder if the *Labrenzia*-*B. minutum* photo is really the best one, as it shows no enrichment at all in the Symbiodiniaceae cell? If possible, I would suggest showing a photo with ¹⁵N enrichment in a Symbiodiniaceae cell.

*We thank the Reviewer for this suggestion and in response have rearranged the panels as suggested and changed the *Labrenzia*-*B. minutum* photos.*

2.23 l. 150-152: it could be because the bacteria were isolated from this culture, and not *Breviolum*, and they are perhaps more specialists than *Labrenzia*?

We agree with this suggestion, and have now added the following text:

L 157-159:

“[...] both Marinobacter sp. and M. aquimarina were more enriched when associated with S. microadriaticum, from which the bacteria were originally isolated (Fig 2, Table S12&S14; ANOVA, FDR <0.001)[...]”

2.24 Figure 3a-b: Presented like this, the results are not super convincing as in (a) all growth curves are all over each other, and in (b) the different starting concentrations give a skewed vision (eg at first glance it could look like EBR in growing better than Untreated). I suggest for each panel to add a small inset with the growth rates from Figure S10 + the associated stats, then it would be very clear very quickly where the changes are.

We thank the reviewer for this suggestion. We have now included the mean specific growth rates and percentage difference to EBR cultures in the figure as well as significance from Table S8, as suggested.

2.25 l. 181: which microalgae? maybe worth mentioning how close they are to Symbiodiniaceae

We have added the microalgae from the references as follows:

L 190:

“IAA is a phytohormone that enhances the growth of plants and recent evidence suggests that it can also promote the growth of microalgae belonging to chlorophytes and diatoms.”

2.26 l. 182: again, which ‘specific bacteria’? if Gammaproteobacteria are known to produce it, it would be worth mentioning.

Many Gammaproteobacteria can produce IAA, we have therefore modified the sentence as follow:

L 192:

“It is synthesised and released by bacteria from multiple classes (including Bacilli, Alpha-, and Gamma-proteobacteria)”

2.27 l. 184: how relevant is 10ug/ml tryptophan compared to what happens in a Symbiodiniaceae culture?

We thank the Reviewer for raising this point. Our decision to use 50 nM tryptophan was to be consistent with the concentration we used for IAA dosing of the Symbiodiniaceae cells. In response to this comment, we attempted to quantify the amount of tryptophan excreted into the surrounding media by both EBR and untreated Symbiodiniaceae via SPE and LC-MS (as described for the IAA analysis). Out of 20 samples, only 2 were above the detection

threshold, but estimated at 53 nM (± 0.004). We therefore believe that the concentration we used is relevant. We have added this justification to the methods:

L 524:

“The final tryptophan concentration of 50 nM was based on SPE and LC-MS analysis (see Extracellular IAA and Tryptophan Extractions and IAA and tryptophan quantification using LC MS/MS below) of EBR and untreated Symbiodiniaceae, which revealed an average exogenous concentration of 53 nM (± 0.004 , n = 2 out of 20 were above the detection threshold).”

L 546:

“Extracellular IAA and Tryptophan Extractions

To quantify the IAA concentrations exuded by the bacterial cells, the supernatant of the bacteria cultures was first filtered through a 0.22 μ m filter to remove all bacteria from the sample. Similarly, to quantify the tryptophan concentrations exuded by Symbiodiniaceae cells, the supernatant of EBR and untreated Symbiodiniaceae cultures were first filtered through a 0.22 μ m filter to remove all cells from the sample. The absence of Symbiodiniaceae and/or bacteria was confirmed via flow cytometry (as above). The SUPELCO manifold and 6 cc 200 mg Oasis HBL cartridge filter columns were used to extract IAA and tryptophan from the samples. [...]”

2.28 l. 190: Table S15 is referred to here, but Tables S12-14 have not yet. Again, some renumbering might be needed.

We thank the Reviewer for highlighting this discrepancy. All Supplementary Table numbers have now been corrected.

2.29 l. 193 and l. 196: Fig 3D (rather than 3C)?

We thank the reviewer for pointing out this error and have changed the figure reference to Fig 3D.

2.30 l. 194: as mentioned above, this drastic difference could be due the the SYmbiodinaiceae cells feeding on the Labrenzia/Marinobacter, rather than internalizing IAA. Please discuss this point.

We thank the reviewer for this suggestion. We believe that we have addressed this point in response to comment 2.3 above.

2.31 Figure 3c-d: in the y-axis legend, please specify that ‘cell’ is ‘bacterial cell’ (could be confused with Symbiodiniaceae cell in panel d). Again, at first glance the results here

are not super striking, and it takes noticing the 10⁻¹¹ vs 10⁻⁷ in the y-axis legends to clearly understand the drastically different levels between Bacterial cultures and co-cultures. I suggest either (i) bringing panel d to the same scale as panel c (10⁻⁷ in the y-axis legend, and the numbers on the y-axis would have an extra 10⁻⁴), or (ii) fusing panels c and d with the same y-axis, and perhaps a y-axis break to be able to show the very low levels clearly.

We have now merged panel c and d of Figure 3 by displaying the data on a logarithmic scale. The resulting panel is clearer, thank to the reviewer's suggestion.

2.32 l. 203: no need to redefine EBR here.

We thank the reviewer for pointing out this error and have removed the text (L208).

2.33 l. 204: how does 50nm IAA compare to the levels measured in the bacterial cultures?

*Our use of 50 nM was chosen based on previously published co-culture work between phytoplankton and bacteria published in Nature (Amin et al. 2015) and on the extraction of IAA excreted by both bacterial species in the presence of tryptophan. We extracted exogenous IAA from approximately 5×10^8 bacterial cells per species at a concentration of 10^7 cells per μL with 50 nM tryptophan, and ran these extracts alongside quantification standards. These results demonstrated an IAA concentration of 0.069 ng/ μL for *Labrenzia* and 0.23 ng/ μL for *Marinobacter* (Table S16) normalised to the same number of cells. We therefore believe that the concentration we used is relevant.*

2.34 l. 206-207: I would quickly discuss here the much less pronounced effect on *B. minutum* (compared to *S. microadriaticum*). In light of the initial correlative analysis showing that IAA did not diminish in the antibiotic-treated *B. minutum*, perhaps the bacteria providing IAA to this culture were not removed by the treatment? Or it relies on other growth hormones?

*We agree that the *B. minutum* had a less pronounced effect relative to *S. microadriaticum*, and it is possible, and very likely, that there are other growth promoting hormones that we did not examine specifically here. We have added the following text:*

L 220:

*“The growth enhancement was less pronounced for *B. minutum* than *S. microadriaticum*, suggesting that other bacterial-derived compounds may further enhance the growth of this Symbiodiniaceae species.”*

2.35 l. 208: ‘specific bacterial partners’, I would remove ‘specific’ because many other bacteria than the ones you studied could be providing it.

We agree with the reviewer and have removed the word ‘specific’ from this sentence (L 220).

2.36 l. 269-270: a sodium hypochlorite wash was mentioned in the results section for the EBR treatment, but not here. Please add it.

We thank the reviewer for pointing out this error and have added the following text:

L 274:

“For the Extracellular Bacteria Removed (EBR) cultures, cultures (n = 5 per Symbiodiniaceae species) were washed with TritonX-100 as above and the detergent supernatant discarded. Cells were resuspended in 20 mL ASW, filtered on a 0.22 µm Durapore membrane filter, and rinsed with 20 mL 6% sodium hypochlorite. Cells were transferred to sterile culture flasks and resuspended in 9 ml ASW+F/2 and treated with antibiotics as above. This process was repeated four times.”

2.37 l. 281: no E culture is shown in this manuscript

We thank the reviewer for pointing out this error and have removed the reference to the E culture (L 284).

2.38 l. 300: please replace ‘16S qPCR’ by ‘16SrRNA gene metabarcoding’

We thank the reviewer for pointing out this error and have changed the text as requested (L 304).

2.39 l. 310: it is unclear to me how ‘relative cell chlorophyll fluorescence’ would give Symbiodiniaceae concentration in a flow cytometry analysis? Didn’t the flow cytometer just count the cells anyway? This needs to be more detailed, ideally with a figure explaining the gating strategies.

We thank the reviewer for bringing this to our attention. This sentence was not correctly written. We have modified it to increase clarity and it now reads:

L 294:

“Symbiodiniaceae cell were identified according to their chlorophyll a fluorescence (650 nm) and subsequently enumerated.”

In addition, we have added a new supplementary figure describing our gating strategy (Fig. S2) see our response to comment 1.5 above.

2.40 l. 316-318: please provide a figure explaining the gating strategies to quantify bacteria.

We have added a figure to the Supplementals (Fig. S2) explaining our gating strategy.

2.41 l. 423: replace ‘16S identification’ by ‘taxonomic identification by 16S rRNA gene sequencing’. Additionally, I am not sure ‘as described above’ is accurate, as the sequences provided in Table S12 are ~1300-1400bp (ie full-length 16S rRNA gene), whereas the primers provided l. 324-326 would only amplify ~400bp. The part l. 420-424 is quite unclear: (i) it starts with Labrenzia as a goal, but then moves to Labrenzia/Marinobacteria/Muricauda; (ii) ‘10 random colonies picked’ how did you end up with 26 colonies then (l. 129)? I would suggest writing its own paragraph for the bacterial isolation (without Labrenzia as a goal), clearly explaining how you isolated them and how you identified them (with the right primers).

We thank the reviewer for this suggestion and have written a separate paragraph for the extractions as suggested:

L 451-460:

“Symbiodiniaceae-associated bacteria isolation

*Bacteria were isolated by plating 10 µL of untreated *S. microadriaticum* and *B. minutum* cultures on 100% Marine Agar (n = 2 per species, BD Difco), incubated at 26°C for 48 h and 10 random colonies picked from each Symbiodiniaceae species (5 from each plate), individually inoculated to sterilised Marine Broth (BD Difco) and incubated at 26°C for 12 h at 180 rpm. This process was repeated until purity. From each inoculation, was extracted as described above and 16S rRNA amplification performed using the primers 27F (5'-AGAGTTTGATCCTGGCTCAG-3') and 1492R (5'-GGTACCTTGTTACGACTT-3')⁶³. From these isolated strains, *Labrenzia* (*Roseibium*) *alexandrii*, *Marinobacter* sp. and *Muricauda aquimarina* with the highest % match (> 98%, E values = 0) were selected for further experiments (Table S12).”*

2.42 l. 427: which nutrient broth was used?

We have corrected the text to:

L 427:

“[...] inoculated to sterilised Marine Broth (BD Difco)”

2.43 l. 433: Tryptophan 50 nM was used in the figures (rather than 10ug/mL), please be consistent.

We thank the reviewer for pointing out this error and have changes the text as requested (as well as elsewhere in the document).

L 188 and 445:

“tryptophan (final concentration 50 nM)”

Reviewer #3 (Remarks to the Author):

This study looks at the dependency of Symbiodiniaceae on their associated bacteria by altering the bacterial composition of two coral endosymbiotic algae species during their free-living stage using detergent and antibiotic treatments. Results show a significant shift in algal cellular metabolism. Specifically, the abundance of monosaccharides and indole-3-acetic acid (IAA) were correlated with the presence of specific bacteria, including members of the Labrenzia and Marinobacter genera.

Single-cell stable isotope tracking revealed that these two bacterial genera are involved in reciprocal exchanges of carbon and nitrogen with Symbiodiniaceae, resulting in significant growth enhancement of the microalga. This work demonstrates how specific bacterial associates influence Symbiodiniaceae fitness.

This is a well written manuscript representing novel data on bacterial/algal associations that fills a significant gap in the literature associated with coral microbiology.

Methods are detailed and appear more than adequate. I have no suggestions. Congratulations to the authors for this contribution.

We sincerely thank reviewer 3 for their time in reviewing our manuscript and for their compliments.

REVIEWER COMMENTS

Reviewer #1 (Remarks to the Author):

I am happy with the edits and new data provided and ready to support the publication of the manuscript. I am especially happy with FigS2, the edits on table S6 and fig S13. I am also glad to see the detailed stats and cytometry data. Lines 281-300 were also extremely helpful in better understanding the protocol applied. Congratulations on the beautiful work.

Reviewer #4 (Remarks to the Author):

The manuscript by Matthews et al., focused on interkingdom metabolic interactions between the coral microalgal symbionts of the family Symbiodiniaceae (autotrophs) and heterotrophic bacteria. The authors used complementary approaches i.e. cultivation based, photophysiology, single cell and metabolite profiling to study the role of bacteria in the growth and function of the algae.

The experiments were conducted using cultures of endosymbiotic algae i.e. free living stages, outside their coral host, and by altering their bacterial composition. Profiling of metabolites coupled to SIP-nanoSIMS approach was used to identify bacterial specific metabolites and exchange of ^{15}N and ^{13}C labelled metabolites between the 2 partners. Generally metabolic interactions between autotrophic algae and bacteria were intensively studied. On a single cell level, using stable isotope labeling and nanoSIMS it has been shown that there is a direct cell to cell metabolic/trophic exchange between the partners. It is therefore no surprise that the autotrophs and heterotrophs exchange metabolites. In the current context however, I see an imperious and immediate benefit to get a deeper understanding of such metabolic relationships due to the importance of the Symbiodiniaceae endosymbionts for the health of reef-building corals. The field is still in its infancy, and I believe such studies are valuable and highly needed. From a methodological perspective, the authors have used a multitude of complementary approaches to address specific questions using experimentally a well-defined system of model organisms and co-cultures. I support the study and appreciate the authors effort; however, I have some major concerns detailed below.

Regarding experiments:

In my opinion there is no clear proof that the extracellular bacteria were removed after the antibiotic & tween treatments.

A simple DAPI or SYBR Green staining, and fluorescence microscopy would have provided a clear evidence.

Algae cells are easily visible based on autofluorescence and bacteria, if any, would have been observed by DAPI. A couple of images showing overlay autofluorescence (green light excitation) with UV for DAPI staining would have been a strong prove in this direction.

Another concern is related to the harsh treatment meant to remove bacteria, how was this influencing algae cell metabolism, or in general growth/ fitness for the downstream co-culturing experiments?

Regarding nanoSIMS data, there is a relatively low number of cells analyzed (20 algae and about 40 bacterial cells), considering that these are pure cultures/co-cultures and not environmental samples, more cells to be analyzed in random fields of view would have not been a problem. Second point, I do not understand why the authors did not calculate/present quantitative data on the exchange rate. How much ^{15}N was transferred to the algae and how much ^{13}C was transferred from the algae to the bacteria, per cell values?

The authors chose to present naked isotopic ratio values. No atom % enrichment or uptake rates that would have clearly shown the exchange rate. Any reason for that?

How was the natural abundance measured? I could not find any description on the analysis on these cells. How many algae and bacteria from unlabeled co-cultures/samples have been measured? There is missing information in the M&M/Supplementary info on these aspects.

Regarding the utilization of the bacteria-derived phytohormone IAA for growth in my opinion a direct prove was not presented. Extracting the IAA from bacteria, growing with labelled ^{13}C and ^{15}N isotopes and feeding directly the algae, followed by the nanoSIMS would have been the direct prove of metabolic dependent IAA utilization by the algae cells. Also a simpler way by using an EA-IRMS one could have measured isotope label from IAA into bulk algae biomass, without single cell resolution but good enough as a direct prove.

Specific comments:

Line 206-207: the fact the IAA was reduced/decreased in the presence of algae is not a prove that was used up by the algae cell for biomass growth. Bacteria could have decreased it's production for unknown reasons or could have been absorbed to the outside EPS or the algae cell. Is there any experimental evidence against the above concerns?

Line 501: How is it possible to have 20 images/areas of 30X30 um for each waver analyzed and have such a low number of 20 algae and 40 bacterial cells as outcome? According to the images presented as representative, there is more than 1 algae per field of view and more than 2 bacterial cells per field of analysis, therefore a discrepancy between the images and claims of analyzed fields of view and the number of cells claimed to be analyzed e.g. 20 algae and 40 bacterial cells.

Line 509: where are these data in atom% presented? Fig 3 presents naked isotope ratios.

Line 511: regarding cell identification - which secondary ion maps or was the secondary electrons or total ion counts images that was used for silhouette/morphological identification of the algae and bacterial cells? It is important to specify details on how cell identification was done. Please show images with ROI definition in the supplementary info.

Line 512: This is not the average enrichment what is presented in fig 3. This figure represents just a comparison between isotopic ratios. For atom percent enrichment calculations see several SIP-nanoSIMS papers from Lawrence Livermore laboratories, MPI Bremen, Uni Vienna, UFZ Leipzig groups.

Line 513: unlabeled samples do not have enrichment! This dot line represents isotopic ratios at natural abundance measured from unlabeled samples (which is not described) and calculated from secondary isotopes of C and N.

REVIEWER COMMENTS

Reviewer #2 (Remarks to the Author):

1.1 The manuscript by Matthews et al., focused on interkingdom metabolic interactions between the coral microalgal symbionts of the family Symbiodiniaceae (autotrophs) and heterotrophic bacteria. The authors used complementary approaches i.e. cultivation based, photophysiology, single cell and metabolite profiling to study the role of bacteria in the growth and function of the algae.

The experiments were conducted using cultures of endosymbiotic algae i.e. free living stages, outside their coral host, and by altering their bacterial composition. Profiling of metabolites coupled to SIP-nanoSIMS approach was used to identify bacterial specific metabolites and exchange of ¹⁵N and ¹³C labelled metabolites between the 2 partners. Generally metabolic interactions between autotrophic algae and bacteria were intensively studied. On a single cell level, using stable isotope labeling and nanoSIMS it has been shown that there is a direct cell to cell metabolic/trophic exchange between the partners. It is therefore no surprise that the autotrophs and heterotrophs exchange metabolites. In the current context however, I see an imperious and immediate benefit to get a deeper understanding of such metabolic relationships due to the importance of the Symbiodiniaceae endosymbionts for the health of reef-building corals. The field is still in its infancy, and I believe such studies are valuable and highly needed. From a methodological perspective, the authors have used a multitude of complementary approaches to address specific questions using experimentally a well-defined system of model organisms and co-cultures. I support the study and appreciate the authors effort; however, I have some major concerns detailed below.

We thank the reviewer for their very positive appraisal of our work.

Regarding experiments:

1.2 In my opinion there is no clear proof that the extracellular bacteria were removed after the antibiotic & tween treatments. A simple DAPI or SYBR Green staining, and fluorescence microscopy would have provided a clear evidence. Algae cells are easily visible based on autofluorescence and bacteria, if any, would have been observed by DAPI. A couple of images showing overlay autofluorescence (green light excitation) with UV for DAPI staining would have been a strong prove in this direction.

We thank the reviewer for their comment. As requested, we have conducted fluorescence microscopy imaging of DAPI-stained cells from the EBR treatments of both Symbiodiniaceae species, further confirming the absence of bacteria cells. We have included the images as a Supplemental figure (Fig. S4) and the methods in the text as follows:

L284-289: To further confirm bacteria absence, fixed EBR cultures (2% glutaraldehyde) were stained with 0.5 mg/mL DAPI for 5 minutes in the dark and filtered on a 0.2 μ m polycarbonate black filter. Cells were then imaged under a Nikon ECLIPSE Ni-L upright fluorescence microscope using a metal halide mercury lamp as a light source, with both DAPI filter (excitation (Ex): 359 nm; emission (Em): 457 nm) for bacteria cell visualisation and Cy5 filter (Ex: 630–650 nm; Em: 660–700 nm) to distinguish Symbiodiniaceae autofluorescence (Figure S4).

1.3 Another concern is related to the harsh treatment meant to remove bacteria, how was this influencing algae cell metabolism, or in general growth/ fitness for the downstream co-culturing experiments?

As mentioned in response to comments 1.12 and 1.16 in the previous review, we selected the antibiotics and concentrations after carefully considering the existing literature. We verified that the antibiotic and detergent treatments we used did not induce photophysiological damage (Fig. S2, Table S7) and Symbiodiniaceae growth, while reduced, was not inhibited (Fig. 3 and Table S2, Lines 94-98 as described below). We also added the following text in response to this concern during the previous set of reviews:

*L94-98: Seven days after this treatment, cell photophysiology remained unchanged for both *S. microadriaticum* and *B. minutum* (Fig. S2, Table S7), but the specific growth rates of *S. microadriaticum* and *B. minutum* were reduced by alteration in the bacterial consortium, by 86.2% and 8.2%, respectively (ANOVA, $p < 0.001$, Table S8), implying growth rate dependence on healthy bacterial consortia.*

L 283:

“We maintained untreated control cultures (i.e., no detergent or antibiotic treatment; $n = 5$) of both Symbiodiniaceae species with their native microbiome alongside the Ab+Tx, EBR and bacteria co-cultures to provide baseline information on Symbiodiniaceae culture health and photophysiological performance.”

1.4 Regarding nanoSIMS data, there is a relatively low number of cells analyzed (20 algae and about 40 bacterial cells), considering that these are pure cultures/co-cultures and not environmental samples, more cells to be analyzed in random fields of view would have not been a problem.

It is important to note that the number of cells quoted by the Reviewer was the minimum per treatment. The average number of cells analysed per treatment was 79.6 ± 8.9 for bacteria and 22.6 ± 1.2 for Symbiodiniaceae (see Table S14). Due to the number of samples processed and the extensive cost of NanoSIMS analyses, we were unfortunately limited in the number of images we were able to acquire per treatment. In addition, after rinsing and drying samples to the wafers, the Symbiodiniaceae cell densities were lower than expected, and some of the images acquired did not contain any Symbiodiniaceae cells. Nevertheless, we aimed to obtain

at least 20 cells for each treatment to ensure the robustness of our analysis. In comparison to other Symbiodiniaceae NanoSIMS studies, the number of Symbiodiniaceae cells imaged ranged from $N = 5$ to 60 symbiont cells per treatment (e.g., Kopp, et al. (2013) Highly dynamic cellular-level response of symbiotic coral to a sudden increase in environmental nitrogen. *MBio*; Rädicker et al., (2018) Using *Aiptasia* as a model to study metabolic interactions in cnidarian-Symbiodinium symbioses. *Frontiers in Physiology*).

1.5 Second point, I do not understand why the authors did not calculate/present quantitative data on the exchange rate. How much ^{15}N was transferred to the algae and how much ^{13}C was transferred from the algae to the bacteria, per cell values? The authors chose to present naked isotopic ratio values. No atom % enrichment or uptake rates that would have clearly shown the exchange rate. Any reason for that? How was the natural abundance measured? I could not find any description on the analysis on these cells. How many algae and bacteria from unlabeled co-cultures/samples have been measured? There is missing information in the M&M/Supplementary info on these aspects.

We thank the reviewer for their comment. Natural abundance was calculated using the abundance of ^{13}C or ^{15}N isotopes in unlabelled cells. We have included further explanation of this in our methods and clarified the replication. As suggested by the reviewer, we have now calculated and changed our ratio values to atom %, and amended Figure 3, Table S14, and the main texts as follows:

L508: Assimilation of the isotope labels (atom % enrichment compared to unlabelled controls) was quantified for a minimum of 20 symbiont cells (^{15}N -labeled) and 40 bacteria cells (^{13}C -labeled) per co-culture by drawing individual regions of interest (ROIs) based on the silhouette of the $^{12}\text{C}^{14}\text{N}$ symbiont or bacteria cells using the ImageJ plugin OpenMIMS (Fig. S5). The measured isotope ratios were converted to ^{15}N or ^{13}C atom fraction (Atom%) as per⁶⁴, which gives the percentage of a specific atom within the total number of atoms, and compared against the baseline from natural isotopic abundances calculated from the Atom% of ^{13}C or ^{15}N isotopes in unlabelled samples (21 unlabelled Symbiodiniaceae and 40 unlabelled bacteria).

1.6 Regarding the utilization of the bacteria-derived phytohormone IAA for growth in my opinion a direct prove was not presented. Extracting the IAA from bacteria, growing with labelled ^{13}C and ^{15}N isotopes and feeding directly the algae, followed by the nanoSIMS would have been the direct prove of metabolic dependent IAA utilization by the algae cells. Also a simpler way by using an EA-IRMS one could have measured isotope label from IAA into bulk algae biomass, without single cell resolution but good enough as a direct prove.

We thank the reviewer for their comment. We would like to highlight that the intention of the study was to investigate whether there was a reciprocal exchange of metabolites between Symbiodiniaceae and associated bacteria. We employed NanoSIMS to provide the first, untargeted evidence of metabolic exchanges in this symbiosis.

The reviewer suggests to extract and purify IAA from the bacterial cells and provide it to the Symbiodiniaceae to ensure its utilisation. However, we have already provided pure IAA to Symbiodiniaceae devoid of extracellular bacteria (see Figure 4), which significantly enhanced the growth of the microalgae. We therefore believe that our data already show the utilisation of IAA by Symbiodiniaceae.

Specific comments:

1.7 Line 206-207: the fact the IAA was reduced/decreased in the presence of algae is not a prove that was used up by the algae cell for biomass growth. Bacteria could have decreased it's production for unknown reasons or could have been absorbed to the outside EPS or the algae cell. Is there any experimental evidence against the above concerns?

We observed a significant increase in Symbiodiniaceae growth response in the presence of IAA, which is a commonly used indicator of IAA utilisation in phytoplankton (see Amin et al. (2015) Interaction and signalling between a cosmopolitan phytoplankton and associated bacteria, Nature) and in plant studies (see Lambrecht et al. (2000) Indole-3-acetic acid: a reciprocal signalling molecule in bacteria–plant interactions, Trends in Microbiology). Nevertheless, we have tempered our language in response to this comment as follows:

L205-209: Our results revealed that extracellular IAA relative abundance was significantly reduced by 1.5×10^4 on average in the presence of Symbiodiniaceae cells compared to the supernatant of the non-tryptophan dosed bacteria, irrespective of the Symbiodiniaceae species (Fig 4d), which suggests that Symbiodiniaceae cells indeed take up bacterial-derived IAA.

In addition, it is important to point out that IAA is a signal molecule affecting gene regulation of photosynthetic organisms but it is not directly used to build biomass (see Teal et al. (2006) Nature Reviews Molecular Cell Biology; Lavy et al. (2016) Development; Tariq & Ahmed (2021) Auxins-Interkingdom Signalling Molecules). Measuring changes in growth rate in response to pure IAA is a well-documented method to measure physiological response to this molecule (e.g., Amin et al., 2015).

1.8 Line 501: How is it possible to have 20 images/areas of 30X30 um for each waver analyzed and have such a low number of 20 algae and 40 bacterial cells as outcome? According to the images presented as representative, there is more than 1 algae per field of view and more than 2 bacterial cells per field of analysis, therefore a discrepancy between the images and claims of analyzed fields of view and the number of cells claimed to be analyzed e.g. 20 algae and 40 bacterial cells.

We thank the reviewer for their comment. As detailed in comment 1.4, we endeavoured to analyse as many cells as possible per image. A low cell density following sample preparation

and high analysis costs limited the number of images we were able to analyse, but we nevertheless achieved a minimum of 20 algae/40 bacteria cells analysed per treatment.

1.9 Line 509: where are these data in atom% presented? Fig 3 presents naked isotope ratios.

We thank the reviewer for their comment. As described in comment 1.5, we have calculated the atom % and updated our methods and figures to reflect this.

1.10 Line 511: regarding cell identification - which secondary ion maps or was the secondary electrons or total ion counts images that was used for silhouette/morphological identification of the algae and bacterial cells? It is important to specify details on how cell identification was done. Please show images with ROI definition in the supplementary info.

We would like to thank the reviewer for their suggestion. We have now included a description of our ROI selection in the methods (as per comment 1.5) and included Figure S5 in support of this.

1.11 Line 512: This is not the average enrichment what is presented in fig 3. This figure represents just a comparison between isotopic ratios. For atom percent enrichment calculations see several SIP-nanoSIMS papers from Lawrence Livermore laboratories, MPI Bremen, Uni Vienna, UFZ Leipzig groups.

We thank the reviewer for their comment. We have calculated the Atom % as suggested, and updated the methods and figures as described in comment 1.5.

1.12 Line 513: unlabeled samples do not have enrichment! This dot line represents isotopic ratios at natural abundance measured from unlabeled samples (which is not described) and calculated from secondary isotopes of C and N.

We thank the reviewer for their comment. We have corrected this error as follows:

L584-586: compared against the baseline from natural isotopic abundances calculated from the Atom% of ^{13}C or ^{15}N isotopes of unlabelled samples (21 unlabelled Symbiodiniaceae and 73 unlabelled bacteria).

REVIEWER COMMENTS

Reviewer #4 (Remarks to the Author):

I am satisfied with the way the authors have revised their manuscript and answered all raised concerns.

There are 2 exceptions, regarding Sx figure, DAPI staining showing prove of no bacteria. The exposure time should be much higher to be able to see dapi stained bacterial cells. As algae are very big, the cell will show a very bright dapi signal and the image max/min will be adjusted accordingly, considering the brighter features in the image.

Thus small features like bacteria of 1-2 um will be peach dark and not observable. I suggest to add some images where algae cells looks saturated/overexposed to indeed show there are no other features/bacteria in the vicinity.

Regrading ROI definition (Fig S5). One can clearly see that drawing of ROIs are not confined to the cell like features (e.g. ROIs 12, 0, 14, 13, 15, 1) and quite often the drawing contour capture either the filter substrate or nearby bacteria. The imprecise drawing of ROIs have implications in the isotopic ratios which consequently are not precisely determined.

Inclusion of filter substrate leads to underestimation of $^{13}\text{C}/^{12}\text{C}$ ratios as the ^{13}C from the cell will get diluted by the ^{12}C predominant in the filter substrate. Moreover, if algae ROIs comprise nearby bacterial cells the isotope ratio values will change either as underestimation or overestimation depending on enrichment/lack of enrichment of the bacterial cells included in the ROI definition of the algae.

Asking the authors to draw again more precise ROIs around cell-like features may not be fully feasible but they should check at least the imprecise defined ROIs and calculate how significantly the isotopic ratio of single cells changed after the precise redrawing and mention this, at least in the SI. I truly hope the isotopic ratios after redrawing ROIs definition for those initially imprecise will not change drastically.

I fully support its publication after meeting the above concerns.

Niculina Musat

REVIEWER COMMENTS

Reviewer #4 (Remarks to the Author):

1.1 I am satisfied with the way the authors have revised their manuscript and answered all raised concerns. There are 2 exceptions, regarding Sx figure, DAPI staining showing prove of no bacteria. The exposure time should be much higher to be able to see dapi stained bacterial cells. As algae are very big, the cell will show a very bright dapi signal and the image max/min will be adjusted accordingly, considering the brighter features in the image. Thus small features like bacteria of 1-2 um will be peach dark and not observable. I suggest to add same images where algae cells looks saturated/overexposed to indeed show there are no other features/bacteria in the vicinity.

We thank the reviewer for their suggestion. We have adjusted the images to overexpose the algal cells and added these images to Figure S4. There remain no visible bacteria in the images.

1.2 Regrading ROI definition (Fig S5). One can clearly see that drawing of ROIs are not confined to the cell like features (e.g. ROIs 12, 0, 14, 13, 15, 1) and quite often the drawing contour capture either the filter substrate or nearby bacteria. The imprecise drawing of ROIs have implications in the isotopic ratios which consequently are not precisely determined. Inclusion of filter substrate leads to underestimation of $^{13}\text{C}/^{12}\text{C}$ ratios as the ^{13}C from the cell will get diluted by the ^{12}C predominant in the filter substrate. Moreover, if algae ROIs comprise nearby bacterial cells the isotope ratio values will change either as underestimation or overestimation depending on enrichment/lack of enrichment of the bacterial cells included in the ROI definition of the algae. Asking the authors to draw again more precise ROIs around cell-like features may not be fully feasible but they should check at least the imprecise defined ROIs and calculate how significantly the isotopic ratio of single cells changed after the precise redrawing and mention this, at least in the SI. I truly hope the isotopic ratios after redrawing ROIs definition for those initially imprecise will not change drastically.

We agree that some of the ROIs could be improved. We have re-examined every NanoSIMS image and found 11 images in which the ROIs could be improved, and thus we have redrawn the ROIs and reanalysed the atom % data in these images. These small adjustments did not impact the statistical significance or interpretation of our results. We have updated the data in the Supplementary and Figure 3 and changed the images in Fig S5 to demonstrate the refined ROI drawings.

1.3 I fully support its publication after meeting the above concerns.

We are very grateful to the reviewer for their input in the manuscript and believe their suggestions have improved the strength of our results and conclusions.

REVIEWERS' COMMENTS

Reviewer #4 (Remarks to the Author):

I am fully satisfied with the final revision of the manuscript.

I have no further comments.

I support the publication of the work.

REVIEWERS' COMMENTS

Reviewer #4 (Remarks to the Author):

I am fully satisfied with the final revision of the manuscript.

I have no further comments.

I support the publication of the work.

We sincerely thank the reviewer for their time in reviewing our manuscript and support of our work.